# Probabilistic Interactive 3D Segmentation with Hierarchical Neural Processes

**Jie Liu** [1]  **Pan Zhou** [2]  **Zehao Xiao** [1]  **Jiayi Shen** [1]  **Wenzhe Yin** [1]  **Jan-Jakob Sonke** [3]  **Efstratios Gavves** [1]

## Abstract

Interactive 3D segmentation has emerged as a promising solution for generating accurate object masks in complex 3D scenes by incorporating user-provided clicks. However, two critical challenges remain underexplored: (1) effectively generalizing from sparse user clicks to produce accurate segmentation and (2) quantifying predictive uncertainty to help users identify unreliable regions. In this work, we propose *NPISeg3D*, a novel probabilistic framework that builds upon Neural Processes (NPs) to address these challenges. Specifically, NPISeg3D introduces a hierarchical latent variable structure with scene-specific and object-specific latent variables to enhance few-shot generalization by capturing both global context and object-specific characteristics. Additionally, we design a probabilistic prototype modulator that adaptively modulates click prototypes with object-specific latent variables, improving the model's ability to capture object-aware context and quantify predictive uncertainty. Experiments on four 3D point cloud datasets demonstrate that NPISeg3D achieves superior segmentation performance with fewer clicks while providing reliable uncertainty estimations. Project Page: `https://jliu4ai.github.io/NPISeg3D_projectpage/`.

## 1. Introduction

Interactive 3D segmentation (Kontogianni et al., 2023; Yue et al., 2023; Valentin et al., 2015; Shen et al., 2020; Zhi et al., 2022; Zhang et al., 2024) seeks to generate precise object masks in complex 3D environments by incorporating user-provided feedback, typically in the form of positive and negative clicks. Positive clicks indicate regions belonging to target objects, while negative clicks define background areas. Through iterative user interaction, models can progressively refine their segmentation predictions, ensuring adaptability and improved accuracy. This inherent flexibility has made interactive 3D segmentation a vital tool for a wide range of real-world applications, including autonomous driving (Ando et al., 2023), robotic manipulation (Zhuang et al., 2023), and augmented reality (Alhaija et al., 2017).

Recent advancements in the field have made substantial progress. For instance, single-object segmentation methods for 3D point clouds (Kontogianni et al., 2023) and multi-object segmentation frameworks (Yue et al., 2023) have significantly pushed the boundaries of interactive 3D segmentation. However, two critical challenges remain insufficiently addressed, hindering the broader adoption and effectiveness of existing methods. The first one is effective few-shot generalization. Interactive 3D segmentation requires models to deliver accurate results with minimal user input. In real-world scenarios, users expect precise segmentation with only a few clicks, requiring models to generalize effectively from sparse user-provided cues. This challenge is amplified in diverse environments with complex scenes and a wide variety of objects, where limited supervision must suffice for reliable segmentation (Schult et al., 2023; Takmaz et al., 2023; Kweon et al., 2024).

The second one is robust uncertainty estimation. The reliability of interactive 3D segmentation models is highly sensitive to variability in user clicks, particularly when input is sparse or ambiguous (Zhou et al., 2023). Uncertainty estimation is critical for interpreting model predictions, such as in model reliability assessment (Wu et al., 2023; Xu et al., 2023; Gong et al., 2023), identifying regions that require further user refinement, and guiding subsequent user interactions. Furthermore, reliable uncertainty quantification is essential in high-stakes applications, such as medical imaging (Rakic et al., 2024) and autonomous driving (Michelmore et al., 2020; Shafaei et al., 2018), where erroneous predictions can have significant consequences. Addressing these challenges is pivotal for advancing the capabilities of interactive 3D segmentation and enabling its effective application across complex real-world scenarios.

**Contributions.** This work proposes NPISeg3D, a novel probabilistic interactive 3D segmentation framework based

[1]University of Amsterdam, Amsterdam, The Netherlands [2]Singapore Management University, Singapore [3]Netherlands Cancer Institute, Amsterdam, The Netherlands. Correspondence to: Jie Liu <j.liu5@uva.nl>, Pan Zhou <panzhou3@gmail.com>.

*Proceedings of the 42nd International Conference on Machine Learning*, Vancouver, Canada. PMLR 267, 2025. Copyright 2025 by the author(s).

on hierarchical neural processes (NPs), to simultaneously address the challenges of few-shot generalization and uncertainty estimation. Our contributions are highlighted below.

Firstly, we introduce the first NP-based framework for interactive 3D segmentation, leveraging the strong few-shot generalization and uncertainty estimation capabilities of NPs (Jha et al., 2022; Garnelo et al., 2018b) as shown in other domains like continual learning (Jha et al., 2024) and semi-supervised learning (Wang et al., 2022; 2023). In our NP segmentation framework, user-provided clicks are treated as the context set providing partial observations about objects of interest, while the remaining unlabeled 3D points in the scene constitute the target set for prediction. Then, our NP framework formulates interactive 3D segmentation as a probabilistic modeling problem, where user-provided clicks (context set) are used to infer object-specific latent variables, and segmentation predictions are generated as a predictive distribution over the target set. This probabilistic formulation enables the model to adaptively update predictions based on user-provided clicks while inherently quantifying uncertainty in its predictions.

Moreover, we develop a hierarchical latent variable structure to further improve the few-shot generalization of our NP segmentation framework. In complex scenes, NP models with a single latent variable often struggle to capture the global structure and adequately model uncertainty (Guo et al., 2023). This challenge is particularly critical in multi-object interactive 3D segmentation setting (Yue et al., 2023) where understanding intricate scene layouts and inter-object relationships is crucial for accurate segmentation. To solve this issue, we introduce a hierarchical latent variable structure with scene-specific and object-specific latent variables. The former models the global scene context and spatial relationships between objects, while the latter, derived from user-provided clicks, captures the fine-grained characteristics of each individual object. This hierarchical design enables the model to effectively integrate global scene understanding with object-aware and click-guided refinement, significantly enhancing its few-shot generalization.

Finally, we propose a probabilistic prototype modulator that dynamically integrates object-specific latent variables into each click prototype, which serves as a localized classifier to guide the segmentation process. By enriching these prototypes with object-aware context, the model derives more informative and adaptive click prototypes, thereby improving its ability to generalize from limited localized user clicks. Furthermore, the probabilistic nature of the modulator, achieved by latent space sampling, enables explicit modeling of prediction uncertainty, offering more reliable and interpretable insights into segmentation confidence.

Extensive experiments on four benchmark datasets demonstrate the superiority of NPISeg3D over state-of-the-arts (SoTAs) in both single- and multi-object interactive 3D segmentation tasks. For instance, on the KITTI-360 dataset, NPISeg3D achieves a significant performance improvement of $8.4\%$ and $4.2\%$ over AGILE3D for single- and multi-object segmentation, respectively. Moreover, unlike existing SoTA approaches such as InterObject3D and AGILE3D which neglect uncertainty estimation, NPISeg3D provides reliable and interpretable uncertainty qualification in model predictions for interactive 3D segmentation.

## 2. Related Work

**Interactive 3D Segmentation.** Despite its importance, interactive 3D segmentation (Valentin et al., 2015; Shen et al., 2020; Zhi et al., 2022; Kontogianni et al., 2020; Yue et al., 2023; Zhang et al., 2024; Zhou et al., 2024) remains relatively underexplored. Pioneering works such as InterObject3D (Kontogianni et al., 2023) and CRSNet (Sun et al., 2023) have established click-based simulation schemes for interactive point cloud segmentation. AGILE3D (Yue et al., 2023) further advances the field by introducing an attention-based model to facilitate interactive multi-object segmentation in 3D point clouds. Additionally, InterPCSeg (Zhang et al., 2024) integrates off-the-shelf semantic segmentation networks to improve their performance using corrective clicks at test-time. Unlike the aforementioned deterministic models, this work introduces the first probabilistic framework for interactive 3D segmentation, providing reliable and insightful uncertainty estimation to guide user interaction.

**Neural Processes.** Neural Processes (NPs) (Garnelo et al., 2018b) learn to approximate stochastic processes by modeling marginal distributions over functions. Conditional Neural Processes (CNPs) (Garnelo et al., 2018a) extend this framework by learning predictive distributions conditioned on context and target sets. Various extensions have been proposed, such as Attentive Neural Processes (ANPs) (Kim et al., 2019), which leverage attention mechanisms for improved representation aggregation, and Transformer Neural Processes (TNPs) (Nguyen & Grover, 2022), which employ transformer architectures to model long-range dependencies. Further advancements focus on enhancing predictive accuracy (Lee et al., 2020), stationarity (Foong et al., 2020), and robustness to noise (Kim et al., 2022). In this work, we extend NPs to interactive 3D segmentation, enabling effective few-shot generalization and uncertainty estimation.

## 3. Preliminary: Neural Processes

Neural Processes (NPs) (Garnelo et al., 2018a;b) are a family of methods designed to approximate the probabilistic distribution over continuous functions conditioned on partial observations. Formally, given a context set $\mathcal{D}_C = (\mathbf{X}_C, \mathbf{Y}_C)$ and a target set $\mathcal{D}_T = (\mathbf{X}_T, \mathbf{Y}_T)$, where

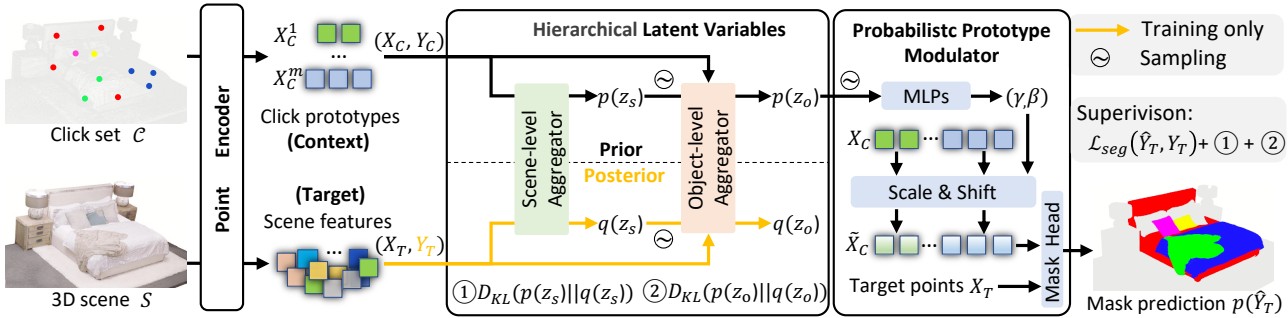

*Figure 1.* **Overview of NPISeg3D.** We formulate interactive 3D segmentation as a probabilistic modeling problem with neural processes. Given a 3D scene $S$ and a user click set $C$, a point encoder encodes them into click prototypes $\mathbf{X}_C$ (context data) and scene features $\mathbf{X}_T$ (target data). Then, we introduce two hierarchical latent variables: scene-level latent variable $\mathbf{z}_s$ and object-level latent variable $\mathbf{z}_o$, to enable probabilistic modeling and capture contextual information across hierarchical levels. In probabilistic prototype modulator, each object-specific latent variable is utilized to generate object-specific weights $(\gamma, \beta)$, which modulate its corresponding click prototypes, thereby enhancing few-shot generalization and providing reliable uncertainty estimation. The posterior distributions of the latent variables are inferred from the target set $(\mathbf{X}_T, \mathbf{Y}_T)$, which supervise the prior during training.

$\mathbf{X}$ and $\mathbf{Y}$ represent input-output pairs, NPs aim to model the conditional distribution $p(\mathbf{Y}_T|\mathbf{X}_T, \mathcal{D}_C)$.

Generally, NPs model the distribution over functions by introducing a global latent variable $\mathbf{z}$, which captures the underlying uncertainty in the function space. This latent variable is sampled from a prior distribution $p(\mathbf{z}|\mathcal{D}_C)$ conditioned on the context set $\mathcal{D}_C$. Given $\mathbf{z}$, the model infers the target ouputs $\mathbf{Y}_T$ based on the conditional distribution $p(\mathbf{Y}_T|\mathbf{X}_T, \mathbf{z})$, where $\mathbf{X}_T$ denotes the target inputs. Thus, the overall NPs model is formulated as:

$$p(\mathbf{Y}_T|\mathbf{X}_T, \mathcal{D}_C) = \int p(\mathbf{Y}_T|\mathbf{X}_T, \mathbf{z})p(\mathbf{z}|\mathcal{D}_C)d\mathbf{z}. \quad (1)$$

Due to the intractable true posterior, previous work resort to amortized variational inference (Kingma & Welling, 2013) to optimize the NPs model. Specifically, let $q(\mathbf{z}|\mathcal{D}_T)$ represent the variational posterior of the latent variable $\mathbf{z}$ inferred from the target set $\mathcal{D}_T$ during training, the evidence lower bound (ELBO) for NPs is derived as:

$$\log p(\mathbf{Y}_T|\mathbf{X}_T, \mathcal{D}_C) \geq \mathbb{E}_{q(\mathbf{z}|\mathcal{D}_T)}\left[\log p(\mathbf{Y}_T|\mathbf{X}_T, \mathbf{z})\right]$$
$$- \mathbb{D}_{\mathrm{KL}}\left[q(\mathbf{z}|\mathcal{D}_T)\|p(\mathbf{z}|\mathcal{D}_C)\right]. \quad (2)$$

The first term of the ELBO is the predictive log-likelihood, which encourages the model generate accurate predictions for the target outputs $\mathbf{Y}_T$. The second term is a Kullback-Leibler (KL) divergence that regularizes the variational posterior $q(\mathbf{z}|\mathcal{D}_T)$ to stay close to the prior $p(\mathbf{z}|\mathcal{D}_C)$. This probabilistic formulation enables the model to generalize from few observed samples (context) to make predictions on unseen data (target), inherently supporting few-shot generalization and uncertainty estimation (Jakkala, 2021; Wang & Van Hoof, 2022; Shen et al., 2023; Wang et al., 2025).

# 4. Methodology

**Task Definition**. Interactive multi-object 3D segmentation operates on a 3D scene represented as a point cloud, denoted by $\mathbf{S} \in \mathbb{R}^{N \times 6}$, where $N$ is the number of 3D points. Each point is characterized by its spatial coordinates $(x, y, z)$ and color features $(r, g, b)$. Then, users provide a sequence of interaction clicks $\mathcal{C} = \{(c_k, o_k)\}_{k=1}^K$, where $c_k = (x_k, y_k, z_k)$ denotes the 3D coordinates of the $k$-th click, $o_k \in \{l\}_{l=0}^M$ is the click label, with $l = 0$ representing the background and $l \in \{1, \ldots, M\}$ denoting the remaining $M$ ($> 1$) objects. Given the 3D scene $\mathbf{S}$ and click sequence $\mathcal{C}$, the goal is to predict the segmentation mask $\mathbf{Y} \in \mathbb{R}^N$, where each element $\mathbf{Y}_i \in \{l\}_{l=0}^M$ denotes the label of the $i$-th point in the scene. Users can iteratively provide corrective clicks to refine segmentation results until they are satisfactory.

## 4.1. NP Framework for Interactive Segmentation

To enable effective few-shot generalization for interactive 3D segmentation with robust uncertainty estimation, we formulate the problem within Neural Processes (NPs) framework, a probabilistic approach. The NPs framework provides a natural formulation for interactive 3D segmentation, where user-provided clicks serve as the context set, and the unclicked points constitute the target set to be predicted. We define the context set and target set in the feature space. As shown in Figure 1, given the 3D scene $\mathbf{S}$ and input clicks set $\mathcal{C}$, a point encoder (Kontogianni et al., 2023; Yue et al., 2023) generates corresponding features $\mathbf{X}_T \in \mathbb{R}^{N \times d}$ for the 3D points and $\mathbf{X}_C = \{\mathbf{X}_C^m\}_{m=0}^M$ for the clicked points. For object $m$, $\mathbf{X}_C^m = \{\mathbf{X}_C^{m,i}\}_{i=1}^{N_C^m}$ is the features of $N_C^m$ clicks with each click feature $\mathbf{X}_C^{m,i} \in \mathbb{R}^d$, and $\mathbf{Y}_C^m = \{\mathbf{Y}_C^{m,i}\}_{i=1}^{N_C^m}$ are corresponding one-hot labels. We refer to $\mathbf{X}_C^m$ as **"click prototypes"** for object $m$, as they serve as click-wise classifiers for segmentation (Yue et al., 2023). Then, the context set is

$\mathcal{D}_C = (\mathbf{X}_C, \mathbf{Y}_C) = \{(\mathbf{X}_C^m, \mathbf{Y}_C^m)\}_{m=0}^M$, while the target set is $\mathcal{D}_T = (\mathbf{X}_T, \mathbf{Y}_T)$. Given the context and target set, NP aims to model the prediction distribution $p(\mathbf{Y}_T | \mathbf{X}_T, \mathcal{D}_C)$.

To effectively model the diverse and complex structures of multiple objects within a 3D scene, we assume that the segmentation functions for each object $m$ are conditionally independent given their respective context information. In interactive segmentation, user clicks are typically provided for distinct objects, which aligns with our assumption of processing each object independently. Specifically, for each object, we introduce an object-specific latent variable $\mathbf{z}_o^m \in \mathbb{R}^d$ to capture fine-grained object-specific characteristics. The latent variable $\mathbf{z}_o^m$ is conditioned on the context set $\mathcal{D}_C^m = (\mathbf{X}_C^m, \mathbf{Y}_C^m)$ for object $m$. Considering that click prototypes $\mathbf{X}_C^m$ inherently encapsulates both feature and label information of clicks, we represent the context set $\mathcal{D}_C^m$ as $\mathbf{X}_C^m$ for simplicity in the following equations. Then, the segmentation function for object $m$ is defined by the following conditional distribution:

$$p(\mathbf{Y}_T^m | \mathbf{X}_T, \mathcal{D}_C^m) = \int p(\mathbf{Y}_T^m | \mathbf{X}_T, \mathbf{z}_o^m)\, p(\mathbf{z}_o^m | \mathbf{X}_C^m)\, d\mathbf{z}_o^m, \tag{3}$$

where $p(\mathbf{z}_o^m | \mathbf{X}_C^m)$ denotes the prior distribution of the latent variable $\mathbf{z}_o^m$, inferred from the context click prototypes $\mathbf{X}_C^m$. $p(\mathbf{Y}_T^m | \mathbf{X}_T, \mathbf{z}_o^m)$ is the predictive distribution conditioned on the latent variable $\mathbf{z}_o^m$, modeling the probability of each point in $\mathbf{X}_T$ belonging to object $m$. By considering multiple objects, the joint conditional distribution is formulated as:

$$p(\mathbf{Y}_T | \mathbf{X}_T, \mathcal{D}_C) = \prod_{m=0}^M \int p(\mathbf{Y}_T^m | \mathbf{X}_T, \mathbf{z}_o^m)\, p(\mathbf{z}_o^m | \mathbf{X}_C^m)\, d\mathbf{z}_o^m. \tag{4}$$

In practice, the prior distribution $p(\mathbf{z}_o^m | \mathbf{X}_C^m)$ is parameterized by a small learnable network. During training, the prior is optimized to approximate the posterior inferred from target data, as described in Eq. (2). This enables the NP framework to effectively capture object-specific characteristics from a limited number of user-provided clicks, facilitating rapid adaptation to unseen objects and enhancing the model's few-shot generalization capability in interactive segmentation. Moreover, the probabilistic formulation in Eq. (4) inherently enables the model to provide uncertainty estimation for segmentation results, offering valuable feedback for guiding user interactions.

Despite its effectiveness, modeling the distribution of functions with only object-level latent variables potentially limits the few-shot generalization capacity of the model (Guo et al., 2023; Shen et al., 2021). This limitation becomes more pronounced in complex scenes, particularly multi-object interactive segmentation settings, where understanding global scene context and inter-object relationships is crucial. To address this challenge, we propose the solution in Sec. 4.2.

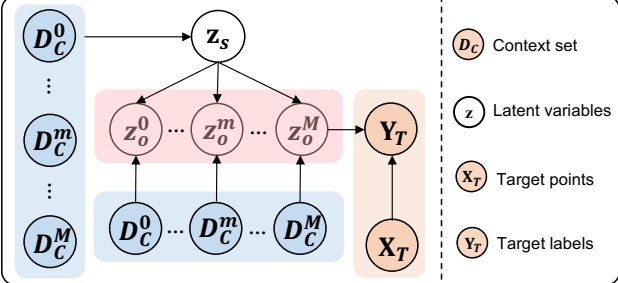

Figure 2. **Computational Graph of our NPISeg3D.** The framework incorporates a hierarchical inference structure with a scene-level latent variable ($z_s$) and an object-level latent variable ($z_o$), capturing contextual information at different spatial levels.

### 4.2. Hierarchical NP Framework for segmentation

To enhance few-shot generalization in complex scenes, we introduce a hierarchical latent variable structure into the NP framework in Eq. (4). As illustrated in Figure 2, this hierarchal structure consists of a scene-specific latent variable $\mathbf{z}_s \in \mathbb{R}^d$ and $M+1$ object-specific latent variable $\mathbf{z}_o^m$. Generally, the scene-specific latent variable $\mathbf{z}_s$ is designed to capture the global scene context and inter-object relationships, providing a holistic understanding of the 3D scene. Simultaneously, the object-specific latent variables $\mathbf{z}_o^m$ focus on encoding fine-grained and object-specific details specified by user-provided clicks. By incorporating the hierarchal latent variables structure, the NP framework in Eq. (4) is reformulated as:

$$p(\mathbf{Y}_T | \mathbf{X}_T, \mathcal{D}_C) = \int \prod_{m=0}^M \left\{ \int p(\mathbf{Y}_T^m | \mathbf{X}_T, \mathbf{X}_C^m, \mathbf{z}_o^m, \mathbf{z}_s) \right.$$
$$\left. \cdot\, p(\mathbf{z}_o^m | \mathbf{z}_s, \mathbf{X}_C^m) d\mathbf{z}_o^m \right\} p(\mathbf{z}_s | \mathbf{X}_C)\, d\mathbf{z}_s, \tag{5}$$

where $p(\mathbf{z}_s | \mathbf{X}_C)$ denotes the prior distribution of the scene-specific latent variable, encoding holistic scene context from all user clicks. $p(\mathbf{z}_o^m | \mathbf{z}_s, \mathbf{X}_C^m)$ models the object-specific latent variable conditioned on the scene-specific latent variable $\mathbf{z}_s$ and click prototypes $\mathbf{X}_C^m$ of object $m$. Lastly, $p(\mathbf{Y}_T^m | \mathbf{X}_T, \mathbf{X}_C^m, \mathbf{z}_o^m, \mathbf{z}_s)$ defines the predictive distribution over the target set for object $m$, and here we introduce an additional condition, i.e., the click prototypes $\mathbf{X}_C^m$, to incorporate click-level details.

This hierarchical design allows the model to seamlessly integrate fine-grained object-specific guidance with global scene understanding, effectively enhancing few-shot generalization and leading to more accurate segmentation. Next, we describe the inference process of these latent variables.

**Scene-Specific Latent Variable.** As illustrated in Figure 1, we infer the distribution of the scene-specific latent variable via a scene-level aggregator. Specifically, we first compute object-level prototypes $\bar{\mathbf{X}}_C^m$ by averaging $N_C^m$ click prototypes $\{\mathbf{X}_C^{m,i}\}_{i=1}^{N_C^m}$ for each object $m$. Then, object-level

prototypes $\{\mathbf{X}_C^m\}_{m=0}^M$ are averaged to construct a scene-level prototype $\bar{\mathbf{X}}_C$. To capture interactions between the scene- and object-level prototypes, we employ a lightweight single-layered transformer (`Trans`), and the scene-specific latent variable is inferred as follows:

$$[\mu_s, \sigma_s] = \texttt{MLP}\big(\texttt{Trans}([\bar{\mathbf{X}}_C; \bar{\mathbf{X}}_C^m])\big), \qquad (6)$$

where $[\mu_s, \sigma_s]$ are the mean and variance of the Gaussian distribution $p(\mathbf{z}_s|\mathbf{X}_C)$, generated using a two-layer multi-layer perceptron (`MLP`). The scene-specific latent variable $\mathbf{z}_s$ captures the global scene context, which is crucial in challenging scenarios with overlapping objects or ambiguous boundaries, where localized guidance alone is insufficient.

**Object-Specific Latent Variables**. For each object $m$, the object-specific latent variable $\mathbf{z}_o^m$ is conditioned on the scene-specific latent variable $\mathbf{z}_s$ and the corresponding object-specific click prototypes $\mathbf{X}_C^m$. This allows $\mathbf{z}_o^m$ to integrate global scene context while adapting to fine-grained and click-specific details. Formally, $\mathbf{z}_o^m$ is generated through an object-level aggregator, as illustrated in Figure 1. This process is formulated as follows:

$$[\mu_o^m, \sigma_o^m] = \texttt{MLP}\Big(\alpha \mathbf{z}_s + (1-\alpha) \sum_{i=1}^{N_C^m} \mathbf{X}_C^{m,i}\Big), \quad (7)$$

where $\mathbf{z}_s$ is a Monte Carlo sample drawn from the prior distribution $p(\mathbf{z}_s|\mathbf{X}_C)$, and $\alpha \in [0,1]$ balances the scene-level context and the object-level context introduced by clicks. The object-specific latent variables $\mathbf{z}_o^m$ effectively capture object-aware context and model object-level uncertainty, enabling robust few-shot generalization.

### 4.3. Probabilistic Prototype Modulator

Click prototypes $\mathbf{X}_C^m$ encapsulate fine-grained and localized cues for segmenting object $m$ and collectively define its decision boundary as an ensemble classifier (Yue et al., 2023). However, each individual click prototype $\mathbf{X}_C^{m,i}$ often lacks awareness of the broader object-level or scene-level context and struggles to effectively capture predictive uncertainty.

To this end, we introduce a probabilistic prototype modulator that dynamically adjusts click prototypes using the corresponding object-specific latent variables $\mathbf{z}_o^m$. These latent variables, derived from our hierarchical modeling approach, capture object-specific uncertainty while integrating global scene context, effectively enriching click prototypes with high-level semantic information. Specifically, the modulated click prototype $\tilde{\mathbf{X}}_C^{m,i,j}$ for object $m$ is defined as:

$$\tilde{\mathbf{X}}_C^{m,i,j} = \gamma(\mathbf{z}_o^{m,j}) \odot \mathbf{X}_C^{m,i} + \beta(\mathbf{z}_o^{m,j}), \qquad (8)$$

where $\mathbf{X}_C^{m,i} \in \mathbb{R}^d$ is the $i$-th click prototype for object $m$, $\mathbf{z}_o^{m,j} \in \mathbb{R}^d$ is the $j$-th Monte Carlo sample drawn from the prior distribution $p(\mathbf{z}_o^m|\mathbf{z}_s, \mathbf{X}_C^m)$, introducing stochasticity

into the model to represent uncertainty. $\gamma(\mathbf{z}_o^{m,j}) \in \mathbb{R}^d$ and $\beta(\mathbf{z}_o^{m,j}) \in \mathbb{R}^d$ are the scale and shift parameters, respectively, computed via a two-layer multi-Layer perceptron (MLP) conditioned on $\mathbf{z}_o^{m,j}$. $\odot$ denotes element-wise multiplication, enabling feature-wise modulation. $\tilde{\mathbf{X}}_C^{m,i,j}$ denotes the modulated click prototype for object $m$ derived from the $i$-th user-provided click and the $j$-th Monte Carlo sample.

By coupling latent variables with click prototype modulation, the framework creates a seamless information flow: `Scene → Objects → Clicks`. This hierarchical structure captures multi-granularity context, including global scene layout, object-specific characteristics, and click-level details, significantly improving few-shot generalization and leading to more accurate segmentation. Moreover, by drawing multiple samples $\mathbf{z}_o^{m,j}$ from the prior distribution, we enable the prototype modulator to be probabilistic, allowing the model to estimate uncertainty in its predictions.

### 4.4. NPISeg3D Pipeline

**Evidence Lower Bound.** To optimize NPISeg3D in Eq. (5), we adopt variational inference (Kingma & Welling, 2013) and derive the evidence lower bound (ELBO) as:

$$\log p(\mathbf{Y}_T|\mathbf{X}_T, \mathcal{D}_C) \geq$$

$$\mathbb{E}_{q(\mathbf{z}_s|\mathbf{X}_T)}\bigg\{ \sum_{m=0}^M \mathbb{E}_{q(\mathbf{z}_o^m|\mathbf{z}_s, \mathbf{X}_T^m)} \log p(\mathbf{Y}_T^m|\mathbf{X}_T, \mathbf{X}_C^m, \mathbf{z}_o^m, \mathbf{z}_s)$$

$$- \mathbb{D}_{\mathrm{KL}}\big[q(\mathbf{z}_o^m|\mathbf{z}_s, \mathbf{X}_T^m)\|p(\mathbf{z}_o^m|\mathbf{z}_s, \mathbf{X}_C^m)\big]\bigg\}$$

$$- \mathbb{D}_{\mathrm{KL}}\big[q(\mathbf{z}_s|\mathbf{X}_T)\|p(\mathbf{z}_s|\mathbf{X}_C)\big], \qquad (9)$$

where $q_\theta(\mathbf{z}_s|\mathbf{X}_T)$ and $q_\phi(\mathbf{z}_o^m|\mathbf{z}_s, \mathbf{X}_T^m)$ denote the variational posteriors of latent variables $\mathbf{z}_s$ and $\mathbf{z}_o^m$, and parameterized by $\theta$ and $\phi$, respectively. The variational posteriors are inferred from the target set $(\mathbf{X}_T, \mathbf{Y}_T)$, and $\mathbf{X}_T^m$ denotes all target point features belonging to object $m$. The prior distributions are supervised by the variational posterior using Kullback–Leibler (KL) divergence, guiding the model to effectively capture richer object-specific information with limited context data and enabling better generalization to unseen target data. See derivations in Appendix A.

**Model Training.** The loss function for NPISeg3D is derived from the ELBO in Eq. (9). In practice, the first term is implemented as a segmentation loss that enforces alignment between predictions and ground truth. The second term consists of two KL divergence regularization terms that constrain the latent variable distributions. Therefore, the overall training objective is formulated as:

$$\mathcal{L} = \mathcal{L}_{seg}(\hat{\mathbf{Y}}_T, \mathbf{Y}_T) + \lambda_{kl}\Big( \mathbb{D}_{\mathrm{KL}}\big[q(\mathbf{z}_s|\mathbf{X}_T)\|p(\mathbf{z}_s|\mathbf{X}_C)\big]$$

$$+ \sum_{m=0}^M \mathbb{D}_{\mathrm{KL}}\big[q(\mathbf{z}_o^m|\mathbf{z}_s, \mathbf{X}_T^m)\|p(\mathbf{z}_o^m|\mathbf{z}_s, \mathbf{X}_C^m)\big]\Big), \quad (10)$$

| Methods | Train→ Eval | IoU@5↑ | IoU@10↑ | IoU@15↑ | Avg↑ | NoC@80↓ | NoC@85↓ | NoC@90↓ | Avg↓ |
|---|---|---|---|---|---|---|---|---|---|
| InterObject3D | | 75.1 | 80.3 | 81.6 | 79.0 | 10.2 | 13.5 | 16.6 | 13.4 |
| InterObject3D++ | ScanNet40→ScanNet40 | 79.2 | 82.6 | 83.3 | 81.7 | 8.6 | 12.4 | 15.7 | 12.2 |
| AGILE3D | (In-domain) | 82.3 | 85.0 | 86.0 | 84.4 | 6.3 | **10.0** | **14.4** | 10.2 |
| **NPISeg3D (Ours)** | | **82.6** | **85.2** | **86.2** | **84.7** | **5.9** | **10.0** | **14.4** | **10.1** |
| InterObject3D | | 76.9 | 85.0 | 87.3 | 83.1 | 6.8 | 8.8 | 13.5 | 9.7 |
| InterObject3D++ | ScanNet40→S3DIS-A5 | 81.9 | 88.3 | 89.3 | 86.5 | 5.7 | 7.6 | 11.6 | 8.3 |
| AGILE3D | (Out-of-domain) | 86.3 | 88.3 | 90.3 | 88.3 | 3.4 | 5.7 | 9.6 | 6.2 |
| **NPISeg3D (Ours)** | | **89.0** | **90.9** | **91.5** | **90.5** | **2.8** | **4.4** | **7.8** | **5.0** |
| InterObject3D | | 73.2 | 83.7 | 87.0 | 81.3 | 7.7 | 10.8 | 18.4 | 12.3 |
| InterObject3D++ | ScanNet40→Replica | 80.4 | 87.5 | 88.8 | 85.6 | 5.9 | 7.5 | 15.7 | 9.7 |
| AGILE3D | (Out-of-domain) | 83.5 | 87.9 | 89.5 | 86.9 | 3.6 | 5.8 | 14.1 | 7.8 |
| **NPISeg3D (Ours)** | | **85.7** | **89.3** | **90.4** | **88.5** | **2.9** | **4.7** | **13.0** | **6.9** |
| InterObject3D | | 10.5 | 22.1 | 31.0 | 21.2 | 19.8 | 19.8 | 19.9 | 19.8 |
| InterObject3D++ | ScanNet40→KITTI-360 | 16.7 | 37.1 | 52.2 | 35.3 | 18.3 | 18.9 | 19.3 | 18.8 |
| AGILE3D | (Out-of-domain) | 40.5 | 44.3 | 48.2 | 44.3 | 17.4 | 18.3 | 18.8 | 18.2 |
| **NPISeg3D (Ours)** | | **44.0** | **48.5** | **52.9** | **48.5** | **16.4** | **17.0** | **17.6** | **17.0** |

*Table 1.* **Quantitative results on interactive multi-object segmentation. Avg** denotes the mean IoU across 5, 10, and 15 clicks, or the mean NoC across 80%, 85%, and 90% IoU thresholds. NPISeg3D is a consistent top-performer, particularly on out-of-domain datasets.

where $\mathcal{L}_{seg}(\hat{\mathbf{Y}}_T, \mathbf{Y}_T)$ is the segmentation loss implemented as the combination of dice loss and cross-entropy loss (Yue et al., 2023; Schult et al., 2023), $\hat{\mathbf{Y}}_T$ denotes the predicted mask for the target point cloud $\mathbf{X}_T$, and $\lambda_{kl}$ is a balancing coefficient for the regularization terms.

**Model Inference.** As shown in Figure 1, given the modulated click prototypes and target points, we generate the final segmentation mask via a mask head. Specifically, we first compute the segmentation logits $\hat{\mathbf{Y}}_T^m$ for object $m$ by aggregating predictions across multiple Monte Carlo samples. This captures the uncertainty introduced by the probabilistic latent variables. Then, a per-point `max` operation is applied to select the most confident response across all user-provided clicks. This process is formulated as:

$$\hat{\mathbf{Y}}_T^m = \max_{i \in \{1,...,N_C^m\}} \frac{1}{N_{z_o}} \sum_{j=1}^{N_{z_o}} \cos(\mathbf{X}_T, \tilde{\mathbf{X}}_C^{m,i,j}), \quad (11)$$

where $\cos(\cdot, \cdot)$ represents the cosine similarity, and $N_{z_o}$ is the number of Monte Carlo samples used to approximate the latent distribution. After computing the logits $\{\hat{\mathbf{Y}}_T^m\}_{m=0}^M$ for $M + 1$ objects (including background), the final segmentation mask $\hat{\mathbf{Y}}_T$ by assigning each point to the object with the highest similarity score. We further obtain the uncertainty map by replacing the mean aggregation in Eq. (11) with their variances, see details in Appendix B.5.

## 5. Experiments

**Datasets.** We follow the dataset setup as in prior work (Yue et al., 2023) and train our model on the ScanNetV2-Train dataset (Dai et al., 2017). For evaluation, we consider two types of datasets: (1) *In-domain dataset*: ScanNetV2-Val (Dai et al., 2017), which shares the same distribution as the training data; (2) *Out-of-domain datasets*: S3DIS (Armeni et al., 2016), collected using different sensors; Replica (Straub et al., 2019), a synthetic indoor dataset;

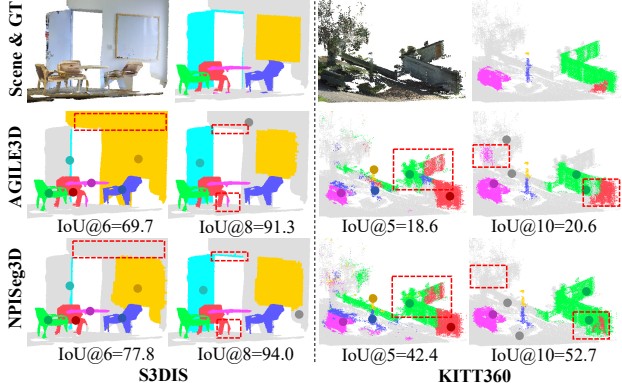

IoU@6=69.7  IoU@8=91.3  IoU@5=18.6  IoU@10=20.6

IoU@6=77.8  IoU@8=94.0  IoU@5=42.4  IoU@10=52.7

**S3DIS**  **KITTI360**

*Figure 3.* **Qualitative results on interactive multi-object segmentation on S3DIS and KITTI360.** Newly added clicks are represented by dark-colored dots. Please zoom in for more details.

and KITTI-360 (Liao et al., 2022), an outdoor LiDAR dataset designed for autonomous driving scenarios. Further details on the dataset setup are provided in Appendix D.1.

**Evaluation Metrics.** We follow prior works (Kontogianni et al., 2023; Yue et al., 2023; Zhang et al., 2024) and evaluate model performance using two key metrics: **(1) NoC@q%** ↓, which measures the average number of clicks required to reach target IoUs of 80%, 85%, and 90%, denoted as NoC@80, NoC@85, and NoC@90. **(2) IoU@k** ↑, which evaluates the average IoU achieved after 5, 10, and 15 clicks, represented as IoU@5, IoU@10, and IoU@15. A maximum of 20 clicks is allowed per instance, and results are averaged across all instances in the multi-object segmentation setting.

**Baselines.** We compare NPISeg3D with three interactive 3D segmentation methods: InterObject3D (Kontogianni et al., 2023), InterObject3D++ (Yue et al., 2023), and AGILE3D (Yue et al., 2023). InterObject3D processes objects sequentially in multi-object scenarios, while InterObject3D++ enhances its performance in multi-object settings with iterative training. AGILE3D adopts an attention-based framework,

| Methods | Train→Eval | IoU@5 ↑ | IoU@10 ↑ | IoU@15 ↑ | Avg ↑ | NoC@80 ↓ | NoC@85 ↓ | NoC@90 ↓ | Avg ↓ |
|---|---|---|---|---|---|---|---|---|---|
| InterObject3D | | 72.4 | 79.9 | 82.4 | 78.2 | 8.9 | 11.2 | 14.2 | 11.4 |
| InterObject3D++ | ScanNet40→ScanNet | 78.0 | 82.9 | 84.2 | 81.7 | 7.7 | 10.0 | 13.2 | 10.3 |
| AGILE3D | (In-domain) | 79.9 | 83.7 | **85.0** | 82.9 | 7.1 | 9.6 | 12.9 | 9.9 |
| **NPISeg3D (Ours)** | | **80.5** | **83.8** | **85.0** | **83.1** | **7.0** | **9.5** | **12.8** | **9.8** |
| InterObject3D | | 72.4 | 83.6 | 88.3 | 81.4 | 6.8 | 8.4 | 11.0 | 8.7 |
| InterObject3D++ | ScanNet40→S3DIS-A5 | 80.8 | **89.2** | **91.5** | 87.2 | 5.2 | 6.7 | 9.3 | 7.1 |
| AGILE3D | (Out-of-domain) | 83.5 | 88.2 | 89.5 | 87.1 | 4.8 | 6.4 | 9.5 | 6.9 |
| **NPISeg3D (Ours)** | | **85.3** | **89.2** | 90.1 | **88.2** | **4.4** | **6.0** | **8.9** | **6.4** |
| InterObject3D | | 64.4 | 80.1 | 86.1 | 76.9 | 8.4 | 10.0 | 12.4 | 10.3 |
| InterObject3D++ | ScanNet40→Replica | 72.6 | 83.8 | 86.7 | 81.0 | 6.9 | 8.1 | 11.2 | 8.7 |
| AGILE3D | (Out-of-domain) | 76.3 | 85.6 | 87.1 | 83.0 | 5.7 | 7.9 | 10.6 | 8.1 |
| **NPISeg3D (Ours)** | | **78.9** | **87.4** | **88.7** | **85.0** | **4.8** | **6.3** | **9.7** | **6.9** |
| InterObject3D | | 14.3 | 26.3 | 35.0 | 25.2 | 19.1 | 19.4 | 19.7 | 19.4 |
| InterObject3D++ | ScanNet40→KITTI-360 | 19.9 | 40.6 | 55.1 | 38.5 | 17.0 | 17.7 | 18.4 | 17.7 |
| AGILE3D | (Out-of-domain) | 44.4 | 49.6 | 54.9 | 49.6 | 14.2 | 15.5 | 16.8 | 15.5 |
| **NPISeg3D (Ours)** | | **55.7** | **57.5** | **60.9** | **58.0** | **11.5** | **13.0** | **14.7** | **13.1** |

*Table 2.* **Quantitative results on interactive single-object segmentation. Avg** denotes the mean IoU across 5, 10, and 15 clicks, or the mean NoC across 80%, 85%, and 90% IoU thresholds. NPISeg3D is a consistent top-performer, particularly on out-of-domain datasets.

| Methods | Datasets | IoU@1 ↑ | IoU@2 ↑ | IoU@3 ↑ |
|---|---|---|---|---|
| InterObject3D | | 38.5 | 54.0 | 62.5 |
| InterObject3D++ | S3DIS | 32.7 | 55.8 | 69.0 |
| AGILE3D | | 58.5 | 70.7 | 77.4 |
| **NPISeg3D** | | **60.5** | **74.0** | **80.0** |
| InterObject3D | | 34.7 | 46.7 | 56.7 |
| InterObject3D++ | Replica | 21.5 | 41.5 | 55.1 |
| AGILE3D | | 53.4 | 63.7 | 67.4 |
| **NPISeg3D** | | **54.6** | **65.7** | **70.3** |
| InterObject3D | | 2.0 | 5.1 | 8.5 |
| InterObject3D++ | KITTI-360 | 3.4 | 7.0 | 11.0 |
| AGILE3D | | 34.8 | 40.7 | 42.7 |
| **NPISeg3D** | | **36.9** | **46.5** | **52.0** |

*Table 3.* **Quantitative results on single-object segmentation task with limited clicks** ($\leq 3$). The best results are highlighted in **bold**.

achieving state-of-the-art (SoTA) performance.

### 5.1. Evaluation on Multi-object Segmentation

**In-domain Results.** As shown in Table 1, NPISeg3D performs competitive and even better than baseline methods on the ScanNet40 dataset by achieving higher segmentation accuracy with fewer user interactions. For example, it attains an IoU@5 of 82.6 and reduces the number of clicks required to achieve 80% IoU (NoC@80) to 5.9, compared to 6.3 for AGILE3D. These results highlight the superiority of NPISeg3D in achieving high-quality interactive segmentation on data distributions similar to the training set.

**Generalization to Out-of-domain.** Notably, NPISeg3D demonstrates strong few-shot generalization capabilities on out-of-domain datasets, including S3DIS-A5, Replica, and KITTI-360, achieving superior segmentation performance with fewer user interactions. As shown in Table 1 (ln. 2), on the S3DIS-A5 dataset, NPISeg3D achieves an impressive 89.0% IoU@5 (+2.7% over AGILE3D) while reducing NoC@80 to 2.8 compared to 3.4 for AGILE3D. Similarly, on the challenging KITTI-360 dataset, it attains 52.9% IoU@5 (+4.7% over AGILE3D) and lowers NoC@90 to 17.6, outperforming AGILE3D's 18.8. These results demon-

| $Z_s$ | $Z_o$ | $M_p$ | S3DIS | | KITTI360 | |
|---|---|---|---|---|---|---|
| | | | mIoU@10 ↑ | NoC@85 ↓ | mIoU@10 ↑ | NoC@85 ↓ |
| - | - | - | 88.7 | 5.5 | 45.4 | 18.1 |
| ✓ | - | ✓ | 90.3 | 5.0 | 46.1 | 17.7 |
| - | ✓ | ✓ | 90.7 | 4.7 | 46.7 | 17.4 |
| ✓ | ✓ | - | 90.4 | 4.7 | 47.8 | 17.2 |
| ✓ | ✓ | ✓ | 90.9 | 4.4 | 48.5 | 17.0 |

*Table 4.* **Ablation study of model components on multi-object segmentation task.** $Z_s$ and $Z_o$ denote the scene-specific and object-specific latent variables, respectively, while $M_p$ represents the probabilistic prototype modulator.

strate the superiority of NPISeg3D in enhancing few-shot generalization on out-of-domain datasets, attributed to our probabilistic framework with hierarchical NPs.

**Qualitative results.** Figure 3 presents the qualitative comparison between AGILE3D and NPISeg3D under the same number of clicks. NPISeg3D predicts more accurate segmentation masks, especially on the challenging KITTI-360 dataset, where it achieves 42.4% IoU@5, significantly outperforming AGILE3D's 18.6% IoU@5. These results underscore NPISeg3D's capability to achieve high-quality segmentation with a limited number of user clicks.

### 5.2. Evaluation on Single-object Segmentation

**Results.** As shown in Table 2, NPISeg3D achieves competitive or superior performance compared to AGILE3D on the ScanNet dataset, attaining an IoU@5 of 80.9% versus AGILE3D's 79.9%. Moreover, NPISeg3D again outperforms most baselines on the out-of-domain datasets. For instance, on KITTI-360, which features complex outdoor LiDAR scenes with substantial domain shifts, NPISeg3D achieves an IoU of 55.7% with 5 clicks, markedly surpassing AGILE3D's 44.4%. These results highlight the superior generalization capability of NPISeg3D in the single-object segmentation setting. See more results in Appendix B.7.

**Low-click Results.** Our method demonstrates remarkable

| Modulation | S3DIS | | KITTI360 | |
|---|---|---|---|---|
| | mIoU@10↑ | NoC@85↓ | mIoU@10↑ | NoC@85↓ |
| ♦Concat | 90.6 | 4.7 | 47.8 | 17.4 |
| ♦Add | 90.4 | 4.9 | 48.2 | 17.2 |
| ◊Ours | 89.6 | 5.0 | 46.1 | 17.8 |
| ♦Ours | 90.9 | 4.4 | 48.5 | 17.0 |

*Table 5.* **Ablation study of prototype modulation strategies.** ♦ and ◊ denote probabilistic and deterministic modulations.

| Method | S3DIS | | KITTI360 | |
|---|---|---|---|---|
| | mIoU@10↑ | NoC@85↓ | mIoU@10↑ | NoC@85↓ |
| MC dropout ($r = 0.1$) | 90.1 | 5.0 | 40.6 | 18.2 |
| MC dropout ($r = 0.3$) | 90.4 | 4.8 | 38.8 | 18.9 |
| MC dropout ($r = 0.5$) | 88.9 | 6.3 | 24.9 | 19.3 |
| **NPISeg3D** ($n = 5$) | 90.8 | 4.5 | 48.6 | 17.0 |
| **NPISeg3D** ($n = 10$) | 90.9 | 4.4 | 48.5 | 17.0 |
| **NPISeg3D** ($n = 20$) | 90.8 | 4.5 | 48.5 | 17.0 |

*Table 6.* **Ablation study of different uncertainty estimation methods with different parameters.** $r$ denotes the dropout rate, and $n$ is the sampling number of latent variables in NPs.

performance in the low-click regime ($\leq 3$ clicks) and out-of-domain scenarios. As shown in Table 3, with only two clicks, NPISeg3D attains IoU scores of 74.0% on S3DIS and 65.7% on KITTI-360. When increased to three clicks, the IoU further improves to 80.0% on S3DIS and 52.0% on KITTI-360, significantly outperforming the state-of-the-art AGILE3D. These results furhter highlight superior few-shot generalization of our NPISeg3D with limited clicks.

## 5.3. Ablation and Analysis

**Effect of Hierarchical Latent Variables.** We conduct an ablation study on the S3DIS and KITTI-360 datasets to assess the impact of the hierarchical latent variables in NPISeg3D. As shown in Table 4, both the scene-specific latent variable $\mathbf{z}_s$ and the object-specific latent variable $\mathbf{z}_o$ contribute to improved segmentation performance. For instance, on S3DIS, introducing $\mathbf{z}_o$ reduces NoC@85 from 5.0 to 4.4 (ln. 2 vs. ln. 6), highlighting its effectiveness in capturing object-specific context. With both $\mathbf{z}_s$ and $\mathbf{z}_o$ (ln. 6), our NPISeg3D achieves the best results, validating the advantage of the hierarchical modeling in capturing multi-granular context and improving few-shot generalization.

**Effect of Probabilistic Prototype Modulator.** As shown in Table 4 (ln. 5 vs. ln. 6), incorporating probabilistic prototype modulator yields notable performance improvements, demonstrating its beneficial effect in improving segmentation accuracy. Furthermore, Table 5 (ln. 3 vs. ln. 4) presents a comparison with a deterministic modulator, where the object-level prototype $\bar{\mathbf{X}}_C^m$ is directly used to generate modulator parameters. The probabilistic approach offers substantial performance gains by capturing uncertainty and enabling more adaptive responses to user inputs. We further

| User | Data Source | $\overline{\text{IoU}}@\overline{3}$↑ | $\overline{\text{IoU}}@\overline{5}$↑ | $\overline{\text{NoC}}@\overline{80}$↓ | $\bar{\text{t}}@\overline{80}$↓ |
|---|---|---|---|---|---|
| Simulator | S3DIS | **84.8** | 92.7 | 2.2 | - |
| Human | | $83.2 \pm 0.6$ | **$93.5 \pm 0.7$** | **$1.7 \pm 0.3$** | 3 min |
| Simulator | KITTI-360 | 79.3 | 86.6 | 3.5 | - |
| Human | | **$81.1 \pm 1.4$** | **$88.1 \pm 0.4$** | **$2.8 \pm 0.4$** | 5 min |

*Table 7.* **User study on annotating 20 objects.** $\bar{\text{t}}@\overline{80}$ represents the time required to reach 80% IoU.

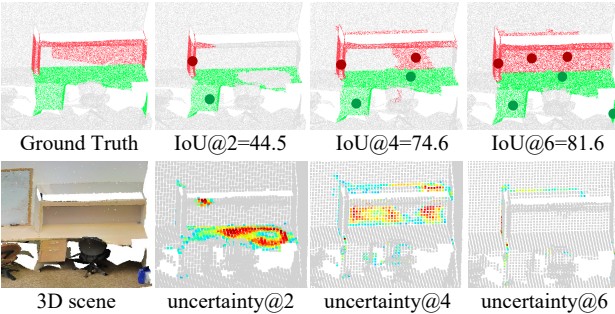

Ground Truth    IoU@2=44.5    IoU@4=74.6    IoU@6=81.6

3D scene    uncertainty@2    uncertainty@4    uncertainty@6

*Figure 4.* **Uncertainty maps of predictions with increasing clicks.** We show mask predictions and uncertainty after $K$ clicks.

explore alternative modulation strategies, such as `Concat` and `Add`, to assess different ways of integrating object-specific latent variables with click prototypes. The results validate the superiority of our method in achieving accurate segmentation with improved generalization.

**Effect of Uncertainty Estimation Strategies.** Table 6 compares MC Dropout (Xiang et al., 2022; Wang et al., 2023) with varying dropout rates ($r = 0.1, 0.3, 0.5$) and NPISeg3D with different latent variable sampling numbers ($n = 5, 10, 20$). Generally, MC Dropout shows a decline in performance as the dropout rate increases, indicating reduced reliability in uncertainty estimation with higher dropout. In contrast, NPISeg3D maintains stable and superior performance across different sampling numbers, demonstrating its robustness and more effective uncertainty modeling. This highlights the superior uncertainty modeling of our NPISeg3D with hierarchical latent variables.

**User Study.** We conducted a user study with real human clicks to evaluate NPISeg3D in practical scenarios. As shown in Table 7, real users achieved performance comparable to the simulator, demonstrating the reliability of our method. Notably, our method consistently achieves superior and robust results across different user behaviors. This improvement is attributed to its uncertainty estimation, which effectively guides user interactions by identifying uncertain regions, reducing the time spent searching for high-error regions. See Appendix C for further details.

**Qualitative Analysis of Uncertainty Estimation.** As shown in Figure 4, NPISeg3D generates reliable and informative uncertainty estimations. Notably, high uncertainty is primarily concentrated in incorrectly segmented regions, along the edges of the predicted mask, and in areas distant from user clicks, reflecting inherently challenging cases in

interactive 3D segmentation. Moreover, uncertainty progressively decreases as more user clicks are provided, indicating the model's increasing confidence in segmentation. For additional examples, see Appendix B.5 and Figure 7 and 8.

## 6. Conclusion

We present NPISeg3D, the first probabilistic framework for interactive 3D segmentation. NPISeg3D formulates the task within the NP framework, incorporating a hierarchical latent variable structure and a probabilistic prototype modulator to enhance the model's capability in few-shot generalization and uncertainty estimation. Experiments demonstrate that NPISeg3D achieves high-quality segmentation and provides reliable uncertainty estimation across diverse 3D scenarios.

**Limitations.** While NPISeg3D demonstrates strong effectiveness, its out-of-domain performance still lags behind in-domain scenarios. Future work could address this by expanding the training set or incorporating domain adaptation techniques to further improve generalization.

## Acknowledgments

This work was partially funded by Elekta Oncology Systems AB and a RVO public-private Partnership grant (PPS2102). Pan Zhou was supported by the Singapore Ministry of Education (MOE) Academic Research Fund (AcRF) Tier 1 grants (project ID: 23-SIS-SMU-070).

## Impact Statement

This work contributes to the field of machine learning by formulating interactive 3D segmentation within a probabilistic framework, enhancing few-shot generalization and uncertainty estimation. Our work has potential applications in areas such as medical imaging, autonomous driving, and robotics. While we do not foresee immediate societal risks, careful deployment is required in high-stakes applications.

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

# A. Derivation of ELBO

The proposed **NPISEG3D** is formulated as:

$$p(\mathbf{Y}_T|\mathbf{X}_T, \mathcal{D}_C) = \int \prod_{m=0}^{M} \left\{ \int p(\mathbf{Y}_T^m|\mathbf{X}_T, \mathbf{X}_C^m, \mathbf{z}_o^m, \mathbf{z}_s)p(\mathbf{z}_o^m|\mathbf{z}_s, \mathbf{X}_C^m)d\mathbf{z}_o^m \right\} p(\mathbf{z}_s|\mathbf{X}_C)\, d\mathbf{z}_s, \tag{12}$$

where $p(\mathbf{z}_s|\mathbf{X}_C)$ and $p(\mathbf{z}_o^m|\mathbf{z}_s, \mathbf{X}_C^m)$ denote the prior distributions of a scene-specific latent variable and each object-specific latent variables, respectively. Considering that $\mathbf{X}_C$ contains both click features and click label information, we substitute $\mathcal{D}_C$ with $\mathbf{X}_C$ on the right-hand side of the above equation and throughout the following derivation. Then, the evidence lower bound is derived as follows:

$$\log p(\mathbf{Y}_T|\mathbf{X}_T, \mathcal{D}_C)$$

$$= \log \int \prod_{m=0}^{M} \left\{ \int p(\mathbf{Y}_T^m|\mathbf{X}_T, \mathbf{X}_C^m, \mathbf{z}_o^m, \mathbf{z}_s)p_\phi(\mathbf{z}_o^m|\mathbf{z}_s, \mathbf{X}_C^m)\, d\mathbf{z}_o^m \right\} p_\theta(\mathbf{z}_s|\mathbf{X}_C)\, d\mathbf{z}_s$$

$$= \log \int \prod_{m=0}^{M} \left\{ \int p(\mathbf{Y}_T^m|\mathbf{X}_T, \mathbf{X}_C^m, \mathbf{z}_o^m, \mathbf{z}_s)\frac{p_\phi(\mathbf{z}_o^m|\mathbf{z}_s, \mathbf{X}_C^m)q_\phi(\mathbf{z}_o^m|\mathbf{z}_s, \mathbf{X}_T^m)}{q_\phi(\mathbf{z}_o^m|\mathbf{z}_s, \mathbf{X}_T^m)}\, d\mathbf{z}_o^m \right\} \frac{p_\theta(\mathbf{z}_s|\mathbf{X}_C)q_\theta(\mathbf{z}_s|\mathbf{X}_T)}{q_\theta(\mathbf{z}_s|\mathbf{X}_T)}\, d\mathbf{z}_s$$

$$\geq \mathbb{E}_{q_\theta(\mathbf{z}_s|\mathbf{X}_T)} \left\{ \sum_{m=0}^{M} \log \int p(\mathbf{Y}_T^m|\mathbf{X}_T, \mathbf{X}_C^m, \mathbf{z}_o^m, \mathbf{z}_s)\frac{p_\phi(\mathbf{z}_o^m|\mathbf{z}_s, \mathbf{X}_C^m)q_\phi(\mathbf{z}_o^m|\mathbf{z}_s, \mathbf{X}_T^m)}{q_\phi(\mathbf{z}_o^m|\mathbf{z}_s, \mathbf{X}_T^m)}\, d\mathbf{z}_o^m \right\} - \mathbb{D}_{\mathrm{KL}}\big[q_\theta(\mathbf{z}_s|\mathbf{X}_T)\|p_\theta(\mathbf{z}_s|\mathbf{X}_C)\big]$$

$$\geq \mathbb{E}_{q_\theta(\mathbf{z}_s|\mathbf{X}_T)} \left\{ \sum_{m=0}^{M} \mathbb{E}_{q_\phi(\mathbf{z}_o^m|\mathbf{z}_s, \mathbf{X}_T^m)} \log p(\mathbf{Y}_T^m|\mathbf{X}_T, \mathbf{X}_C^m, \mathbf{z}_o^m, \mathbf{z}_s) - \mathbb{D}_{\mathrm{KL}}\big[q_\phi(\mathbf{z}_o^m|\mathbf{z}_s, \mathbf{X}_T^m)\|p_\phi(\mathbf{z}_o^m|\mathbf{z}_s, \mathbf{X}_C^m)\big] \right\}$$

$$- \mathbb{D}_{\mathrm{KL}}\big[q_\theta(\mathbf{z}_s|\mathbf{X}_T)\|p_\theta(\mathbf{z}_s|\mathbf{X}_C)\big],$$

where $q_\theta(\mathbf{z}_s|\mathbf{X}_T)$ and $q_\phi(\mathbf{z}_o^m|\mathbf{z}_s, \mathbf{X}_T^m)$ are variation posteriors of their corresponding latent variables, $\theta$ and $\phi$ are parameters of inference modules for $\mathbf{z}_s$ and $\mathbf{z}_o^m$. This ELBO effectively decomposes the log-likelihood into three terms: (1) a reconstruction term, which maximizes the expected log-likelihood of the predictions $\mathbf{Y}_T^m$ under the latent variables; (2) a KL divergence term for each object-specific latent variable $\mathbf{z}_o^m$, which aligns the approximate posterior with the prior; and (3) a KL divergence term for the scene-specific latent variable $\mathbf{z}_s$, ensuring that the global context is appropriately captured. This formulation explicitly models uncertainty at both the scene and object levels, enabling the framework to effectively handle complex 3D segmentation tasks with both global and localized user interactions.

# B. Additional Results.

## B.1. Part Segmentation Results.

| Methods | IoU@5 ↑ | IoU@10 ↑ | IoU@15 ↑ | NoC@80 ↓ | NoC@85 ↓ | NoC@90 ↓ |
|---|---|---|---|---|---|---|
| InterObject3D | 61.9 | 70.2 | 73.2 | 14.5 | 16.7 | 18.2 |
| InterObject3D++ | 60.0 | 70.8 | 72.6 | 13.9 | 16.4 | 17.8 |
| AGILE3D | 59.6 | 70.1 | 74.4 | 14.5 | 16.7 | 18.0 |
| **NPISeg (Ours)** | **63.2** | **73.2** | **77.0** | **13.0** | **15.2** | **17.3** |

*Table 8.* **Quantitative results on part-wise segmentation using the PartNet dataset.** The results are reported for multi-part segmentation tasks, where the models are trained on ScanNet40. The best-performing results are highlighted in **bold**. Our model consistently outperforms others in part segmentation accuracy.

To further evaluate the effectiveness of our method in part-wise segmentation, we conduct additional experiments on PartNet under the multi-part interactive segmentation setting, where multiple parts of an object must be segmented simultaneously.

For evaluation, we select six of the most common objects with multiple parts, namely chairs, beds, scissors, sofas, tables, and lamps. To ensure diversity in the evaluation set, we randomly sample ten point clouds for each object category. This

setup allows us to assess the generalization ability of our model across different object geometries and part compositions. Since such selection involves randomness, we plan to release the object ids for a fair comparison with our method.

Table B.1 presents the quantitative results on the PartNet dataset for multi-part segmentation tasks. Our proposed NPISeg3D consistently outperforms existing methods, achieving the highest IoU across all evaluation thresholds (IoU@5, IoU@10, and IoU@15), demonstrating its superior segmentation accuracy. Notably, NPISeg3D requires fewer user clicks to reach the desired IoU targets, as evidenced by the lowest NoC@80, NoC@85, and NoC@90 values. These results indicate that our hierarchical probabilistic modeling effectively captures fine-grained part-level details while maintaining global scene context, addressing the limitations of prior approaches like AGILE3D, which struggled with part-wise segmentation. The improved performance highlights the robustness of NPISeg3D in handling complex multi-part segmentation scenarios with minimal user interaction. The qualitative results in Figure 6 further demonstrate effectiveness of our framework.

### B.2. Benefits of Incorporating Previous Mask.

| Methods | Datasets | IoU@5 ↑ | IoU@10 ↑ | IoU@15 ↑ | NoC@80 ↓ | NoC@85 ↓ | NoC@90 ↓ |
|---|---|---|---|---|---|---|---|
| w/o Prev. Mask | S3DIS | **89.0** | 90.5 | 91.0 | 2.9 | 4.6 | 7.8 |
| w/ Prev. Mask | | **89.0** | **90.9** | **91.5** | **2.8** | **4.4** | **7.8** |
| w/o Prev. Mask | KITTI360 | 43.5 | 47.8 | 52.3 | 16.6 | 17.3 | 17.8 |
| w/ Prev. Mask | | **44.0** | **48.5** | **52.9** | **16.4** | **17.0** | **17.6** |

*Table 9.* **Benefit of incorporating previous mask into our framework.** Results are reported for multi-object segmentation tasks, with models trained on the ScanNet40 dataset. Better results are highlighted in **Bold**.

In our approach, we leverage the previous segmentation mask to enhance the interactive segmentation process by incorporating historical predictions into the current interaction. Specifically, the previous mask is first converted into a one-hot representation, capturing class-wise information at each point. This one-hot encoding is then processed through a learnable mask encoder, implemented as a four-layer Multi-Layer Perceptron (MLP), which outputs a refined representation of size N×15. The encoded mask features are subsequently concatenated with the point-wise features, providing enriched contextual information that facilitates more accurate predictions in subsequent interactions.

The results in Table 9 validate the effectiveness of incorporating the previous mask in both the S3DIS and KITTI360 datasets. The inclusion of previous mask information leads to consistently improved IoU scores across different click counts, demonstrating the model's ability to leverage past segmentation knowledge for better refinement. Moreover, the number of clicks required to reach target IoU thresholds (NoC metrics) is significantly reduced, highlighting the efficiency gains achieved through this strategy. These results affirm that utilizing prior segmentation masks effectively guides the model towards more precise and efficient interactive segmentation.

### B.3. Ablation on Click Simulation Strategy during Training

| Methods | Datasets | IoU@5 ↑ | IoU@10 ↑ | IoU@15 ↑ | NoC@80 ↓ | NoC@85 ↓ | NoC@90 ↓ |
|---|---|---|---|---|---|---|---|
| Iterative | S3DIS | **89.3** | **91.1** | **91.7** | 2.9 | 4.5 | **7.8** |
| RITM | | 89.0 | 90.9 | 91.5 | **2.8** | **4.4** | **7.8** |
| Iterative | KITTI360 | **44.5** | 46.7 | 50.5 | 17.1 | 17.6 | 18.3 |
| RITM | | 44.0 | **48.5** | **52.9** | **16.4** | **17.0** | **17.6** |

*Table 10.* **Comparison of our model with different click simulation strategies during training.** Results are reported for multi-object segmentation tasks, with models trained on the ScanNet40 dataset. Better results are highlighted in **Bold**.

AGILE3D (Yue et al., 2023) demonstrated the effectiveness of iterative training for interactive multi-object segmentation. However, iterative training incurs high computational costs, making it less practical for large-scale applications. In contrast, we adopt the more efficient RITM sampling strategy (Sofiiuk et al., 2022), which balances training efficiency and segmentation performance. See more analysis in Sec. D.2.

Table 10 compares our method using Iterative and RITM click simulation strategies during training on the S3DIS and KITTI360 datasets. For the S3DIS dataset, the Iterative method achieves slightly higher IoU scores across all click counts,

with IoU@5 reaching 89.3 compared to RITM's 89.0. However, RITM exhibits a lower NoC@80 (2.8 vs. 2.9), suggesting that it requires fewer clicks to achieve the same segmentation quality at the 80% IoU threshold. Both methods perform similarly in NoC@85 and NoC@90.

On the KITTI360 dataset, which poses greater challenges due to its outdoor LiDAR environment, the RITM strategy significantly outperforms the Iterative method in IoU@10 and IoU@15, with improvements of 1.5 and 2.4 points, respectively. Additionally, RITM achieves better NoC performance, indicating its superior efficiency in complex scenarios.

Overall, while the Iterative method slightly excels in high-quality indoor segmentation (S3DIS), RITM demonstrates better generalization to challenging outdoor environments (KITTI360) by achieving higher IoU with fewer user interactions. Considering the balance between computation efficiency and performance, we adopt RITM as the click sampling strategy during our model training.

### B.4. Computation Analysis

| Model | Params/MB | FLOPs/G | Train Speed/days | Inference Speed/ms |
|---|---|---|---|---|
| Inter3D | 37.86 | 0.47 | - | 80 |
| Inter3D++ | 37.86 | 0.47 | - | 80 |
| AGILE3D | 39.30 | 4.73 | 6.4 | 60 |
| **NPISeg3D** | 40.00 | 4.94 | 2.7 | 65 |

*Table 11.* **Computational analysis of NPISeg and state-of-the-art (SOTA) methods.** The traing and inference speed are evaluated on a single NVIDIA A6000 GPU.

Table 11 presents a comparative analysis of NPISeg3D and state-of-the-art methods in terms of model size, computational complexity, and efficiency. NPISeg3D has a slightly larger parameter size (40.00 MB) compared to AGILE3D (39.30 MB) and Inter3D variants (37.86 MB), primarily due to the inclusion of the scene-level and object-level aggregators, as formulated in Eq. (6) and Eq. (7). Despite the increased FLOPs (4.94G), NPISeg3D achieves a significantly faster training time of 2.7 days, outperforming AGILE3D's 6.4 days, attributed to the efficient random and iterative training strategy (see more details in Sec. D.2). At inference, NPISeg3D runs at 65 ms per scan—slightly slower than AGILE3D (60 ms) but substantially faster than Inter3D (80 ms). This demonstrates that NPISeg3D balances the added complexity of probabilistic modeling with competitive runtime efficiency, making it well-suited for real-time applications.

Overall, these results confirm that NPISeg3D achieves a favorable balance between computational efficiency and segmentation performance, demonstrating its potential in practical applications.

### B.5. Uncertainty Estimation

In our probabilistic segmentation framework, uncertainty estimation plays a crucial role in guiding user interactions and improving model reliability. To achieve this, we replace the mean operation used in segmentation logits computation (Eq. 11) with variance, leveraging multiple Monte Carlo samples to quantify the model's confidence. Specifically, our approach estimates uncertainty by measuring the variability across different latent samples. Since multiple click classifiers exist for each object, we further aggregate uncertainty by selecting the maximum value across all clicks, ensuring that the regions with the highest ambiguity are highlighted for further refinement.

Formally, the uncertainty map for object $m$ at each point is computed as:

$$\mathbf{U}_T^m = \max_{i \in 1, \ldots, N_C^m} \left( \frac{1}{N_{z_o}} \sum_{j=1}^{N_{z_o}} \left( \cos(\mathbf{X}_T, \tilde{\mathbf{X}}_C^{m,i,j}) - \mathbb{E}[\cos(\mathbf{X}_T, \tilde{\mathbf{X}}_C^{m,i,j})], \right)^2 \right) \tag{13}$$

where $\mathbf{U}_T^m$ denotes the uncertainty map for object $m$, $N_C^m$ represents the number of click prototypes for object $m$, and $\mathbb{E}[\cdot]$ denotes the expectation over Monte Carlo samples. The variance quantifies the spread of predictions across different latent samples, while the `max` operation identifies the highest uncertainty value among all click prototypes. The overall uncertainty map $\mathbf{U}_T$ is obtained by taking the maximum uncertainty across all object classes, including the background, ensuring a

comprehensive representation of segmentation confidence across the entire scene. Formally, it is computed as:

$$\mathbf{U}_T = \max_{m \in \{0,...,M\}} \mathbf{U}_T^m.$$ (14)

This aggregated uncertainty map helps identify the most uncertain regions, guiding user interactions for more efficient refinement. This probabilistic formulation provides an interpretable measure of segmentation confidence, allowing the model to identify regions requiring further corrective interactions. Such an approach enhances the interactive segmentation process by focusing user efforts on the most uncertain areas, thereby improving the overall segmentation quality in complex multi-object 3D environments.

We provide visualization examples of uncertainty maps in Figure 7 and Figure 8.

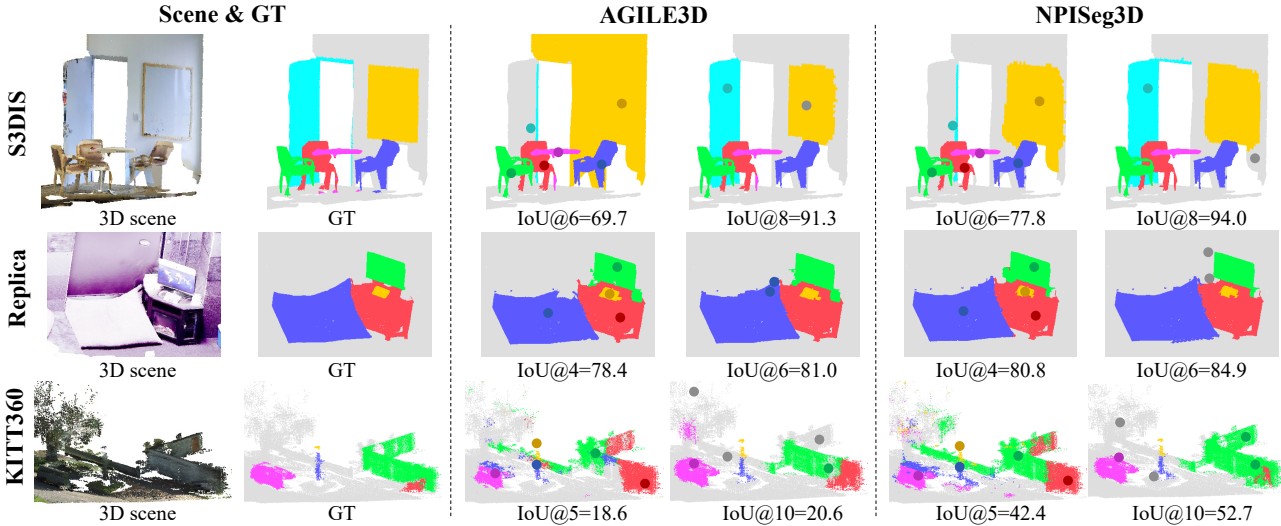

Figure 5. **Qualitative comparison between the state-of-the-art (SoTA) method AGILE3D and our proposed NPISeg3D on the interactive multi-object segmentation task.** Newly added clicks are represented by dark-colored dots. Our NPISeg3D consistently achieves higher IoU scores with the same number of clicks, particularly on the challenging outdoor LiDAR dataset KITTI-360.

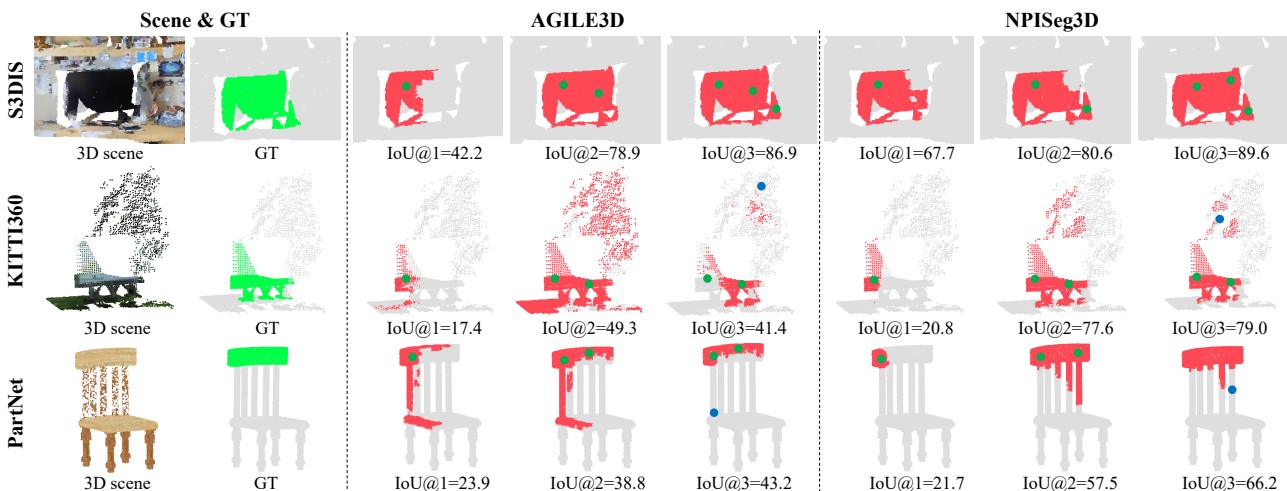

Figure 6. **Qualitative comparison between the state-of-the-art (SoTA) method AGILE3D and our proposed NPISeg3D on the interactive single-object segmentation task.** Newly added clicks are represented by dark-colored dots. Our NPISeg3D consistently achieves higher IoU scores with the same number of clicks. Notably, our NPISeg3D also achieve decent segmentation performance on part-wise segmentation on the PartNet dataset.

## B.6. Qualitative Comparison with SoTA method.

In Figure 5 and Figure 6, we provide additional comparisons between the state-of-the-art (SoTA) method AGILE3D and our proposed NPISeg3D on the multi-object and single-object segmentation tasks, respectively. Generally, our NPISeg3D consistently achieves higher IoU scores than AGILE3D with the same number of clicks across diverse datasets. The improvement is particularly pronounced on the challenging outdoor LiDAR dataset KITTI-360. For example, in Figure 5, our NPISeg3D achieves an IoU of 42.4 after 5 clicks, significantly outperforming AGILE3D's 18.6. These results further demonstrate the strong few-shot generalization derived from the NP framework with hierarchical latent variable modeling.

## B.7. More Qualitative Results.

We present more qualitative results regarding segmentation performance and uncertainty estimation in Figure 7 and Figure 8. As shown in these figures, our method not only demonstrates strong few-shot generalization capability—i.e., generating accurate segmentation masks with minimal user input—but also provides reliable and meaningful uncertainty estimation. Notably, high uncertainty is concentrated around erroneous regions and object boundaries. This insight suggests that uncertainty maps could be leveraged to guide the selection of click candidates for subsequent interaction rounds. We leave this exploration for future work.

# C. User Study.

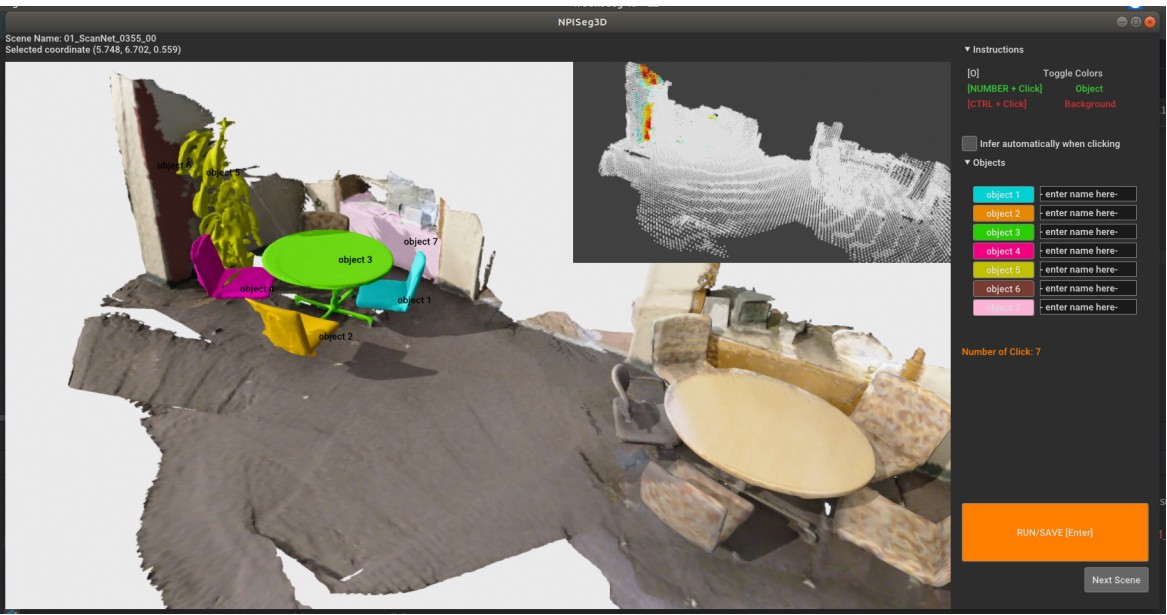

*Figure 9.* **User interface for interactive segmentation with integrated uncertainty estimation as guidance.** In the main window, users provide input clicks, and the model generates a segmentation prediction. The upper-right window displays uncertainty estimation for the current segmentation, guiding users on where to focus in the next interaction round.

To move beyond simulated user clicks and evaluate performance with real human interactions, we conduct user studies.

**User Interface.** To this end, we extend the user-friendly interface from (Yue et al., 2023) to incorporate uncertainty estimation. As shown in Figure 9, the enhanced interface consists of two windows: a main window where users provide clicks and receive corresponding segmentation predictions, and a sub-window in the upper-right corner that visualizes uncertainty estimation for the current segmentation, guiding users on where to focus in the next interaction round. The software is cross-platform and browser-compatible, supporting both interactive single- and multi-object segmentation. To enhance user interaction, various keyboard shortcuts are designed for efficiency. For example, Ctrl + Click designates a background region, while Number + Click specifies an object. We appreciate the open-source tool provided by (Yue et al., 2023) and will release our code, along with the improved annotation tool, to facilitate future research.

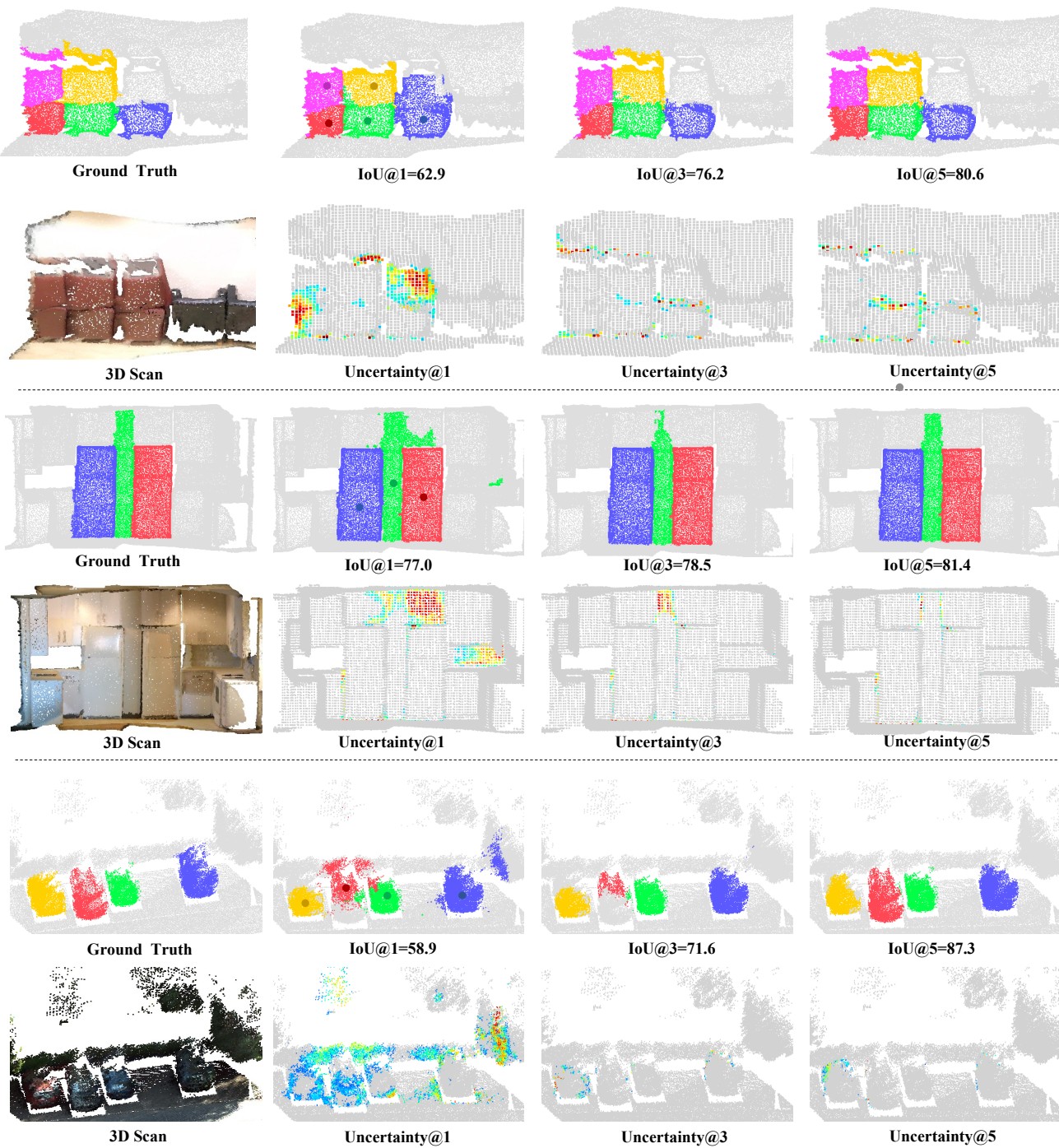

*Figure 7.* **Qualitative Results on Interactive Single-Object Segmentation.** In the multi-object segmentation task, IoU@k denotes the average Intersection over Union (IoU) after k clicks per object. For clarity, only the first click per object is visualized, with subsequent clicks omitted. The uncertainty map highlights regions of model uncertainty, where brighter areas correspond to higher uncertainty. Notably, high uncertainty is predominantly localized to object boundaries and regions distant from user-provided clicks.

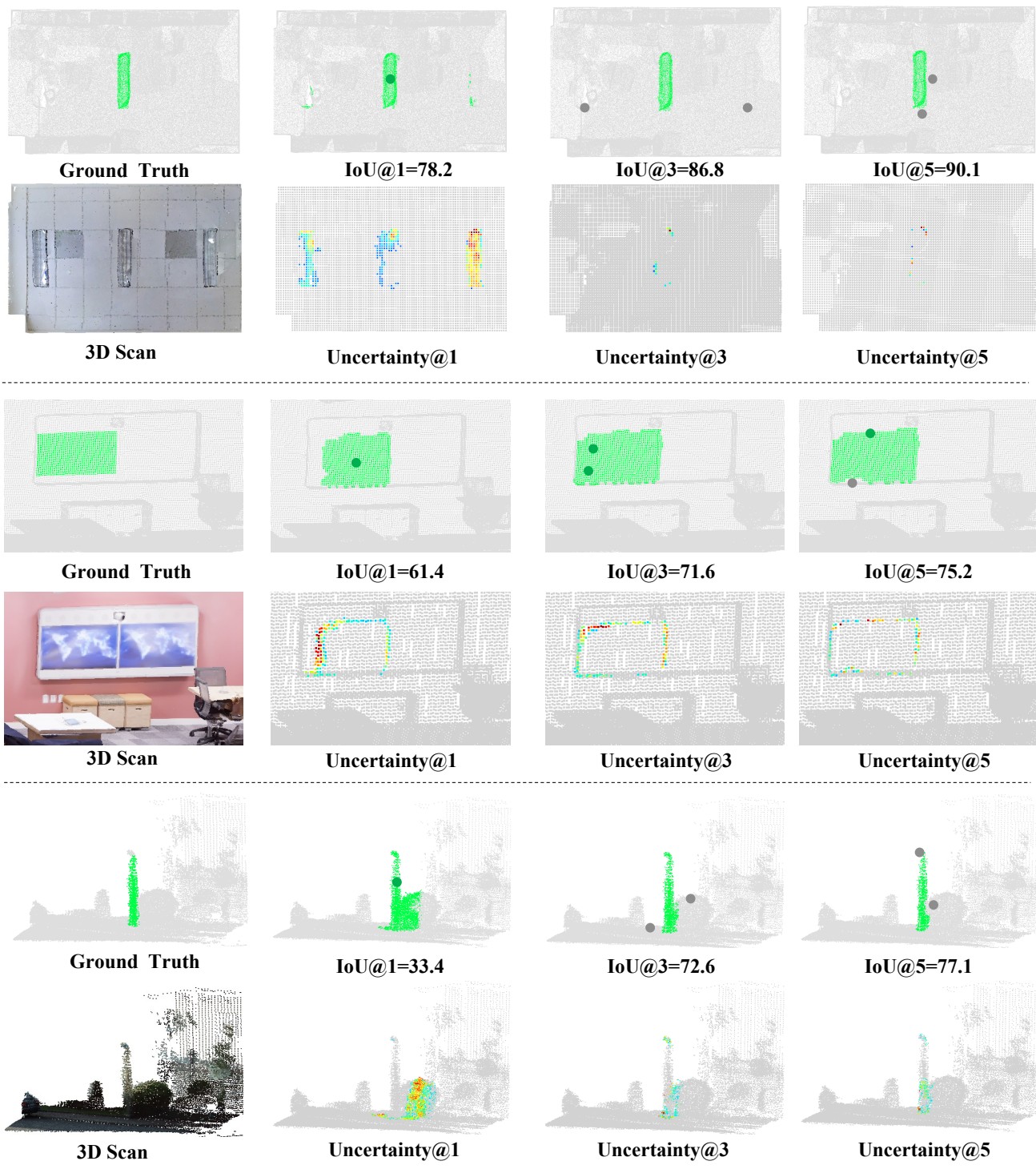

*Figure 8.* **More qualitative results on the interactive single-object segmentation task.** In multi-object segmentation task, IoU@k denotes the average IoU after k clicks per object. We show the first click per object, and omit following clicks for better visualization. In the uncertainty map, brighter regions indicate higher uncertainty. Our method provides reliable uncertainty estimation, with high uncertainty primarily focused on object edges and areas far from user clicks.

**Design.** We conducted a user study involving 10 participants with no prior experience with point cloud annotation. Firstly, each participant was as assigned the task to annotate 4 scenes, where two of the scenes from the S3DIS dataset and the other two from the KITTI-360 dataset. Each of these scenes consists of 4-7 objects, which cover various categories, including chairs, tables, cars, etc. Participants were allowed to refine the segmentation results step by step based on their own preferences, rather than being constrained to selecting the regions with the highest error as the next click candidate, as done in simulation. Secondly, during annotation, each participant continued annotating until they were satisfied with the results. Throughout the process, we recorded the average intersection-over-union (IoU) score and the number of clicks. Finally, we computed the average of these metrics across all participants and present the results in Table 7.

## D. Experimental Setup Details.

### D.1. Dataset Details.

We adopt the experimental setting from prior work (Yue et al., 2023) and train our model on the ScanNetV2-Train dataset (Dai et al., 2017), which contains 1,200 indoor scenes. Evaluation is conducted across four diverse datasets, covering both indoor and outdoor environments: ScanNetV2-Val (Dai et al., 2017), S3DIS (Armeni et al., 2016), Replica (Straub et al., 2019), and KITTI-360 (Liao et al., 2022). Additionally, we further evaluate the part-level segmentation capability of our NPISeg3D and existing models on the PartNet dataset (Mo et al., 2019).

ScanNetV2 (Dai et al., 2017) consists of richly annotated indoor scenes, serving as the primary dataset for training and validation. S3DIS (Armeni et al., 2016) features 272 indoor scenes across six large areas, presenting a significant domain gap from ScanNetV2. Replica (Straub et al., 2019) is a photorealistic indoor dataset comprising high-quality 3D reconstructions of real-world environments, offering diverse scene layouts and materials for testing model generalization. KITTI-360 (Liao et al., 2022), an outdoor LiDAR point cloud dataset, is designed for 3D perception tasks in autonomous driving and provides large-scale outdoor scenes.

PartNet (Mo et al., 2019) focuses on part-level segmentation, offering hierarchical annotations of 3D objects. This dataset challenges the model's ability to perform fine-grained segmentation by decomposing objects into semantic subcomponents. These datasets collectively provide a diverse and comprehensive benchmark to evaluate the robustness and generalizability of our model across varying domains and tasks.

### D.2. Implementation Details.

**Detailed Model Setting**. In NPISeg3D, the point encoder consists of a backbone network and an attention network to effectively process 3D point cloud data. Specifically, we adopt the Minkowski Res16UNet34C (Choy et al., 2019) backbone, following prior works (Kontogianni et al., 2023; Yue et al., 2023; Schult et al., 2023). The 3D scene is first quantized into $N'$ spare voxels with a fixed resolution of 5cm, ensuring efficient and consistent representation, as done in previous studies (Kontogianni et al., 2023; Yue et al., 2023; Schult et al., 2023). The backbone processes the sparse voxelized input and outputs a feature map of size $N' \times 96$, which is further projected to 128 channels by a $1 \times 1$ convolution. Following (Yue et al., 2023), we employ an attention network composed of multiple layers that facilitate interaction between click and scene features. These layers include click-to-scene, scene-to-click, and click-to-click attention mechanisms, enabling effective fusion of user-provided inputs with the global scene context to enhance segmentation performance.

**Model Training**. We train NPISeg3D end-to-end for 600 epochs using the Adam optimizer with an initial learning rate of 0.0005. The learning rate is reduced by a factor of 0.1 after 500 epochs to facilitate convergence. Training is performed on a single Tesla A6000 GPU with a batch size of 5. The KL loss coefficient $\lambda_{kl}$ in Eq. (10) is set to 0.005. For the segmentation loss $\mathcal{L}_{seg}$, we use a combination of cross-entropy loss and dice loss, with coefficients of 1 and 2, respectively.

**Click Simulation Strategy for Training.** In interactive 3D segmentation, click simulation during training plays a crucial role in aligning the model's behavior with real-world user interactions. The multi-object iterative training strategy proposed in (Yue et al., 2023) has been demonstrated to effectively improve segmentation performance by simulating user clicks iteratively based on the model's predictions from previous rounds. This iterative approach closely mimics the test-time click sampling strategy, enhancing the model's ability to refine segmentation with progressive user feedback.

However, despite its effectiveness, this iterative simulation strategy is computationally expensive and time-consuming (Sofiiuk et al., 2022), as evidenced by the runtime analysis in Table 11. The high computational cost makes it impractical for large-scale training scenarios. In contrast, interactive 2D segmentation commonly employs a more efficient strategy, known

as Random and Iterative Training Mechanism (RITM) (Sofiiuk et al., 2022), which strikes a balance between training efficiency and performance. RITM initializes the training process by randomly sampling an initial set of clicks, followed by a limited number of iteratively sampled clicks to refine segmentation predictions.

Inspired by RITM, we extend its application to the interactive 3D segmentation task, including the multi-object segmentation setting. Specifically, our approach combines random and iterative sampling strategies to optimize both efficiency and accuracy. First, an initial set of user clicks is randomly sampled, providing diverse training samples across different objects. Then, a variable number of iterative clicks, ranging from 0 to $N_{iter}$, are added progressively based on the model's prediction errors, ensuring better alignment with real-world interaction scenarios.

**Click Simulation Strategy for Inference.** Following prior works (Kontogianni et al., 2023; Yue et al., 2023; Zhang et al., 2024), we adopt a standardized automated click simulation strategy to ensure reproducible evaluation. Specifically, the first click is placed at the center of the object within the ground truth mask. Subsequent clicks are iteratively placed at the centroid of the largest misclassified region by comparing the predicted segmentation with the ground truth mask. This process continues until the segmentation reaches the maximum click limit is reached, i.e., totally 20 clicks.

