# OpenReview forum: "Probabilistic Interactive 3D Segmentation with Hierarchical Neural Processes"
_ICML.cc/2025/Conference — ICML 2025 poster_

### Official Review · Reviewer_uwh8 · 2025-03-08

**Overall Recommendation:** 3

**Summary:**

This paper addresses the problem of interactive 3D segmentation, where the model segments target objects based on positive and negative user clicks. This paper proposes a probabilistic framework built upon Neural Processes (NPs) to enhance model generalisation. Specifically, the model aggregates object embeddings into scene-level embeddings to capture global context, and subsequently updates object embeddings to represent object-specific characteristics. The method is evaluated on several benchmarks and demonstrates improved generalisation.

**Claims And Evidence:**

Yes.

**Essential References Not Discussed:**

N/A

**Experimental Designs Or Analyses:**

The method demonstrates improved generalization on the benchmarks. However, the paper lacks sufficient architecture details for reproduction and full understanding. The authors should provide code to clarify these details.

Additionally, the method achieves only minor improvements on the ScanNet benchmark; the authors should discuss and analyze the reasons behind this.

**Methods And Evaluation Criteria:**

The method is evaluated on commonly used benchmarks.

**Other Comments Or Suggestions:**

N/A

**Other Strengths And Weaknesses:**

N/A

**Questions For Authors:**

Please refer to the comments above.

**Relation To Broader Scientific Literature:**

The method leverages a probabilistic framework, which may benefit other related tasks that require generalization capabilities.

**Theoretical Claims:**

N/A

---

> ### Author Rebuttal · Authors · 2025-03-29
>
> ***Q1: Architecture details for reproduction and code for clarification.***
>
> Thanks for your valuable suggestions. In Appendix D.2 (pp. 19), we provided additional architectural details of our framework. Specifically, following AGILE3D, the point encoder in Figure 1 consists of a backbone—Minkowski Res16UNet34C—along with an attention module for click-scene interaction. The architectural components related to Neural Processes (NPs) are depicted in Figure 1 and detailed in Sec. 4.2–4.4, including details about scene-level and object-level aggregators (Sec.4.2), probabilistic prototype modulator (Sec.4.3), and training objectives (Sec.4.4).
>
> To better illustrate the hierarchical NP structure, we further provided a graphical model at the following anonymous link: https://github.com/anonusers2025/rebuttal_figure/blob/main/graphical_model.pdf. This figure offers a more visual and accessible explanation of how our model integrates scene-level and object-level latent variables for probabilistic interactive 3D segmentation.
>
> ***We will further improve  the description about architecture details in the final version*** to facilitate reproduction and understanding. ***We will also release our code*** to facilitate implementation and future research.
>
> ***Q2: Discuss and analyze the reasons behind minor improvements on the ScanNet.***
>
> We thank the reviewer for the comment. Our model is exclusively trained on ScanNet, and we observe relatively modest improvements over AGILE3D on this benchmark. This is expected for several reasons.
>
> First, all baseline models and ours are trained on ScanNet, and these models have already seen a large number of structurally similar indoor scenes during training, the segmentation task becomes relatively easy in this in-domain setting. As a result, strong baselines like AGILE3D already perform very well with only a few user clicks, e.g., reaching 82.3% IoU after 5 clicks, and the performance quickly saturates. Under such conditions, the benefits of our probabilistic design are less pronounced, and the potential for further improvement is naturally limited.
>
> Second, our method is specifically designed to handle uncertainty and model generalization through a probabilistic framework with hierarchical latent variables. These advantages are less evident in in-domain settings such as ScanNet, where the model benefits from exposure to similar scenes during training. In contrast, they become significantly more valuable when applied to challenging, unseen, and out-of-domain scenarios. For example, on KITTI-360, which features unstructured outdoor LiDAR scans with large domain gaps, NPISeg3D achieves +10.9% and +9.8% mIoU improvement over AGILE3D under single-object and multi-object settings, respectively. Similar trends are observed on S3DIS and Replica.
>
> These results highlight that ***while our method shows modest gains in in-domain settings like ScanNet, it delivers substantial benefits under more realistic, ambiguous, and out-of-domain conditions—where generalization and uncertainty modeling are crucial.***

---

### Official Review · Reviewer_E5kU · 2025-03-13

**Overall Recommendation:** 3

**Summary:**

Main Contributions & Findings
The paper introduces NPISeg3D, a novel probabilistic framework for interactive 3D segmentation, leveraging Hierarchical Neural Processes (NPs) to tackle two key challenges:
1. Few-shot generalization – enabling accurate segmentation from sparse user clicks.
2. Uncertainty estimation – providing reliable confidence measures to guide user interactions.
Key findings include:
* NPISeg3D achieves superior segmentation performance with fewer user clicks compared to state-of-the-art (SoTA) baselines.
* It improves generalization in both in-domain and out-of-domain settings.
* The probabilistic framework enables explicit uncertainty quantification, enhancing interpretability.
Main Algorithmic/Conceptual Ideas
1. Hierarchical Neural Processes (NPs):
    * Introduces a scene-specific latent variable (for capturing global scene context) and object-specific latent variables (for modeling fine-grained object characteristics).
    * These hierarchical latent variables improve generalization and facilitate probabilistic modeling of segmentation tasks.
2. Probabilistic Prototype Modulator:
    * Dynamically adjusts click-based segmentation prototypes using learned object-specific latent variables.
    * Improves adaptability to user interactions and enhances uncertainty estimation.
3. Probabilistic Formulation:
    * The model treats user clicks as context data and remaining 3D points as target data in a probabilistic setting.
    * Uses variational inference to estimate segmentation probabilities and model uncertainties.
4. Efficient Training & Inference:
    * The model employs variational inference with an evidence lower bound (ELBO) to optimize segmentation accuracy while maintaining robust uncertainty estimates.
    * At inference, segmentation masks are generated using a Monte Carlo-based probabilistic approach.
Main Results
* Quantitative Performance:
    * NPISeg3D outperforms AGILE3D, InterObject3D, and other SoTAs on multiple 3D segmentation benchmarks, including ScanNet, S3DIS, Replica, and KITTI-360.
    * In out-of-domain settings, NPISeg3D consistently achieves higher IoU (Intersection over Union) with fewer user interactions.
    * Reduces the number of clicks needed to achieve high accuracy (e.g., reducing NoC@80 on KITTI-360 from 17.4 (AGILE3D) to 16.4).
* Qualitative & User Study Findings:
    * NPISeg3D provides more precise segmentation masks with fewer clicks.
    * Generates uncertainty maps that highlight unreliable regions, guiding further user interaction.
    * Real-user experiments confirm that the model effectively improves annotation efficiency and accuracy.

**Claims And Evidence:**

yes

**Essential References Not Discussed:**

No

**Experimental Designs Or Analyses:**

yes

**Methods And Evaluation Criteria:**

Yes

**Other Comments Or Suggestions:**

NO

**Other Strengths And Weaknesses:**

Strengths
1. Originality & Novelty
    * The paper presents the first probabilistic framework for interactive 3D segmentation, which is a significant departure from deterministic methods like AGILE3D and InterObject3D.
    * The hierarchical neural process (NP) formulation is an innovative adaptation of NPs to segmentation tasks, particularly in an interactive, few-shot setting.
    * The probabilistic prototype modulator is a novel mechanism that dynamically refines click prototypes, improving both segmentation performance and uncertainty estimation.
2. Significance & Impact
    * The proposed method has strong potential applications in real-world domains such as autonomous driving, robotic perception, and medical imaging, where segmentation reliability is critical.
    * The integration of uncertainty quantification in an interactive framework could fundamentally change user interaction strategies in 3D annotation, making segmentation more adaptive and efficient.
    * Demonstrates strong few-shot generalization capabilities, making it highly applicable in low-data scenarios, a common challenge in real-world applications.
3. Empirical Rigor & Comprehensive Evaluation
    * The paper evaluates NPISeg3D across four benchmark datasets (ScanNet, S3DIS, Replica, KITTI-360), including both in-domain and out-of-domain settings, demonstrating its robustness.
    * The quantitative comparisons against strong baselines (InterObject3D, AGILE3D) show consistent superiority in terms of IoU and click efficiency (NoC@80, NoC@85, NoC@90).
    * The user study strengthens the claim that NPISeg3D enhances annotation efficiency and provides practical benefits in real-world interactive segmentation tasks.
4. Clarity & Reproducibility
    * The paper is generally well-written with clear explanations of the methodology, probabilistic formulation, and hierarchical latent variable design.
    * The detailed ablation studies (evaluating latent variables, modulation strategies, uncertainty modeling, etc.) provide insights into why each component is effective.
    * Mathematical formulations are rigorous and systematically derived, making the theoretical contributions accessible to the reader.

Weaknesses
1. Limitations in Out-of-Domain Generalization
    * Despite its strong performance, NPISeg3D still lags behind in-domain performance when applied to out-of-domain datasets, such as KITTI-360.
    * The paper does not explore domain adaptation techniques, which could further improve generalization to unseen datasets.
2. Computational Complexity & Scalability
    * The probabilistic inference framework introduces additional computational overhead compared to deterministic models like AGILE3D.
    * The need for Monte Carlo sampling for latent variable inference could slow down real-time interaction, particularly in large-scale datasets.
    * The scalability of NPISeg3D to very large point clouds (e.g., in high-resolution LiDAR-based perception) remains unclear.
3. Clarity & Accessibility of Technical Concepts
    * While mathematically rigorous, the paper assumes a high level of familiarity with Neural Processes (NPs), which may limit accessibility for non-experts.
    * The explanation of the hierarchical NP structure could be improved with more intuitive visualizations or concrete examples.

Overall, I believe this paper is highly valuable, making significant contributions to the field of interactive 3D segmentation through its novel probabilistic framework. The weaknesses I have mentioned are not fundamental flaws but rather aspects where further discussion could enhance the clarity, applicability, and impact of the work.

**Questions For Authors:**

Please see my detailed comments above

**Relation To Broader Scientific Literature:**

The key contributions of NPISeg3D build upon and extend several areas of research in interactive 3D segmentation, probabilistic modeling, and neural processes. Below is a structured discussion of how its contributions relate to the broader scientific literature.

1. Interactive 3D Segmentation
Related Work:
* Interactive segmentation has been explored in both 2D and 3D domains, with works like InterObject3D (Kontogianni et al., 2023) and AGILE3D (Yue et al., 2023) focusing on multi-object segmentation in point clouds.
* CRSNet (Sun et al., 2023) used click-based simulation to refine segmentation masks iteratively, but lacked probabilistic uncertainty modeling.
* SemanticPaint (Valentin et al., 2015) introduced interactive 3D labeling using real-time feedback but relied on handcrafted features rather than learning-based models.
NPISeg3D's Novelty:
* Unlike prior deterministic models, NPISeg3D is the first to introduce a probabilistic framework into interactive 3D segmentation.
* It improves few-shot generalization, reducing the number of user clicks required for accurate segmentation.
* Uncertainty estimation is incorporated, which was missing in prior interactive segmentation methods.


2. Few-shot Learning & Neural Processes
Related Work:
* Neural Processes (NPs) (Garnelo et al., 2018) introduced a probabilistic approach to function approximation, learning to model distributions over functions with minimal supervision.
* Conditional Neural Processes (CNPs) (Garnelo et al., 2018) extended NPs by conditioning outputs on observed context points.
* Attentive Neural Processes (ANPs) (Kim et al., 2019) improved information aggregation by integrating attention mechanisms.
* NP-based approaches have been used in continual learning (Jha et al., 2024) and semi-supervised learning (Wang et al., 2023) but had not yet been applied to interactive segmentation.
NPISeg3D's Novelty:
* It formulates interactive segmentation as a probabilistic function approximation problem, leveraging hierarchical neural processes to improve generalization from limited user inputs.
* The hierarchical latent variable structure (scene-level and object-level latent variables) extends standard NPs, making them more effective for structured segmentation tasks.
* It introduces a probabilistic prototype modulator, which enhances adaptability to new objects in few-shot scenarios.

3. Uncertainty Estimation in Segmentation
Related Work:
* Uncertainty estimation has been widely studied in Bayesian Deep Learning (Gal & Ghahramani, 2016) and Monte Carlo Dropout (MC Dropout) (Xiang et al., 2022).
* Deep Gaussian Processes (DGPs) (Jakkala, 2021) modeled uncertainty in deep learning by capturing distributions over functions.
* Uncertainty-aware methods have been applied in medical imaging (Rakic et al., 2024) and autonomous driving (Michelmore et al., 2020), where error quantification is crucial.
* Existing segmentation models such as AGILE3D (Yue et al., 2023) and InterPCSeg (Zhang et al., 2024) neglected uncertainty estimation.
NPISeg3D's Novelty:
* It directly incorporates predictive uncertainty into the segmentation pipeline, allowing users to identify unreliable regions.
* Unlike MC Dropout, which samples from model weights, NPISeg3D models structured uncertainty via latent space sampling.
* It outperforms MC Dropout-based approaches, as shown in ablation studies.


4. Multi-object and Multi-modal Segmentation
Related Work:
* Multi-object segmentation approaches like AGILE3D (Yue et al., 2023) used attention-based mechanisms for segmenting multiple objects.
* OpenMask3D (Takmaz et al., 2023) introduced open-vocabulary segmentation in 3D, but lacked interactivity.
* PointSAM (Zhou et al., 2024) proposed prompt-based 3D segmentation, leveraging large vision models, though it does not incorporate user feedback.
NPISeg3D's Novelty:
* Unlike AGILE3D, which relies on deterministic attention mechanisms, NPISeg3D introduces hierarchical latent variables to model inter-object relationships probabilistically.
* Unlike OpenMask3D, NPISeg3D is not restricted to pre-defined object categories and instead adapts dynamically to user inputs.

5. Interactive Machine Learning & Human-in-the-Loop AI
Related Work:
* Human-in-the-loop (HITL) learning has been used in areas like active learning (Xu et al., 2023) and iterative annotation frameworks (Sofiiuk et al., 2022).
* SemanticPaint (Valentin et al., 2015) allowed users to refine 3D segmentations iteratively.
* Interactive semi-supervised segmentation (Wang et al., 2022) explored how user inputs could improve segmentation accuracy.
NPISeg3D's Novelty:
* It integrates HITL learning with probabilistic uncertainty modeling, enabling more efficient human-in-the-loop correction.
* Unlike existing interactive segmentation frameworks, it prioritizes regions with high uncertainty, guiding user inputs more effectively.

**Theoretical Claims:**

yes

---

> ### Author Rebuttal · Authors · 2025-03-29
>
> ***Q1: Limitations in Out-of-Domain Generalization.  (1) Out-of-domain performance falls short in-domain. (2) The paper does not explore domain adaptation techniques.***
>
> Thank you for your insightful comments. Similar to previous methods like AGILE3D, our model is trained solely on ScanNet and evaluated on out-of-domain datasets such as KITTI-360 to assess its generalization capability. **Due to the significant domain gap between ScanNet (indoor RGB-D scenes) and KITTI-360 (outdoor LiDAR scenes), segmentation on KITTI-360 remains inherently challenging.** As a result, even SOTA method like AGILE3D achieves only 44.4% mIoU after 5 clicks under the single-object setting.
>
> Nevertheless, NPISeg3D achieves substantial improvements over previous state-of-the-art (SoTA) methods on KITTI-360. Specifically, it improves mIoU by 10.9% and 9.8% over AGILE3D under the single-object and multi-object settings, respectively, demonstrating the robustness of our approach even under large distribution shifts.
>
> To further enhance out-of-domain performance, one feasible solution is to consider domain adaption techniques, e.g, domain-specific fine-tuning. As shown in the table below, **our model—when fine-tuned on KITTI-360—achieves significantly better performance across all metrics, approaching levels that are practical for downstream tasks.** For example, mIoU@5 improves from 44.0% to 79.2%, and the number of clicks (NoC@90) required to reach 90% IoU decreases from 17.6 to 12.3. These results demonstrate the strong adaptability of our method to specific domains when fine-tuning data is available.
>
> |               | **mIoU@5** | **mIoU@10** | **mIoU@15** | **NoC@80** | **NoC@85** | **NoC@90** |
> |---------------|------------|-------------|-------------|------------|------------|------------|
> | **w/o fine-tuning** | 44.0       | 48.5        | 52.9        | 16.4       | 17.0       | 17.6       |
> | **w/ fine-tuning**  | 79.2       | 82.8        | 85.4        | 8.5        | 10.8       | 12.3       |
>
> We agree that domain adaptation is a valuable and complementary direction for interactive 3D segmentation,  and  will consider more advanced domain adaptation techniques such as parameter-efficient fine-tuning and test-time adaption in future research.
>
> ***Q2: Computational Complexity & Scalability.***
>
> As shown in Table 11 of the Appendix, our method introduces only marginal computational overhead due to its probabilistic modeling. **The parameter size increases by just 1.8% (40.00MB vs. 39.30MB), and the FLOPs remain comparable (4.94G vs. 4.73G).** Although Monte Carlo sampling with 5 samples adds some overhead, both sampling and decoding are fully parallelized. As a result, our method achieves an inference speed of 65 ms per forward pass, closely matching AGILE3D’s 60 ms and supporting real-time interaction.
>
> Moreover, our iterative and random training strategy accelerates training by approximately 2.5× compared to AGILE3D, further highlighting the scalability of our approach for large-scale datasets.
>
> **Despite this slight overhead, our method delivers significant performance improvements**. For instance, on the high-resolution LiDAR dataset KITTI-360, it surpasses AGILE3D by 11.3% mIoU with just 5 clicks under the single-object setting, demonstrating that **the modest computational cost is well justified by the substantial performance gains.**
>
> ***Q3: Clarity & Accessibility of Technical Concepts. (1) Rigorous math of Neural Processes may limit accessibility for non-experts. (2). The explanation of the hierarchical NP structure could be improved.***
>
> Thank you for the helpful suggestion. We fully agree that it is important to make the mathematical formulation more intuitive and accessible, especially for readers who are less familiar with Neural Processes (NPs). **In the final version, we will revise the exposition to improve clarity and incorporate more intuitive explanations where appropriate.**
>
> ***To better illustrate the hierarchical NP structure, we further provided a graphical model at the following anonymous link:*** https://github.com/anonusers2025/rebuttal_figure/blob/main/graphical_model.pdf. This figure offers a more visual and accessible explanation of how our model integrates scene-level and object-level latent variables for probabilistic interactive 3D segmentation. We will further improve and refine this part in the final version to enhance readability and understanding.

---

### Official Review · Reviewer_TzH1 · 2025-03-13

**Overall Recommendation:** 4

**Summary:**

This paper presents NPISeg3D, a novel probabilistic framework for interactive 3D segmentation based on neural processes (NPs), which addresses the key challenges of generalizing from sparse user clicks and quantifying predictive uncertainty. The framework introduces a hierarchical latent variable structure and a probabilistic prototype modulator to enhance few-shot generalization and provide reliable uncertainty estimation.

**Claims And Evidence:**

Yes.

**Essential References Not Discussed:**

Most of the relevant literature has already been discussed.

**Ethical Review Concerns:**

No ethical concerns

**Experimental Designs Or Analyses:**

Comprehensive comparative experiments are carried out with existing methods such as InterObject3D, InterObject3D++, and AGILE3D, demonstrating the superiority of NPISeg3D in segmentation accuracy and user interaction efficiency.

**Methods And Evaluation Criteria:**

The paper conducts experiments on multiple datasets, including ScanNetV2, S3DIS, Replica, and KITTI-360, covering both indoor and outdoor environments, ensuring a certain degree of representativeness.

**Other Comments Or Suggestions:**

The current model takes point clouds as input. I hope users can discuss the applications of other 3D representations in the article, such as 3D Gaussian Splatting (3DGS). Specifically, whether it is possible to achieve interactive 3DGS segmentation and the potential integration with open-vocabulary 3DGS semantic segmentation methods, such as  GOI [1] and Chatsplat [2].


[1] Goi: Find 3d gaussians of interest with an optimizable open-vocabulary semantic-space hyperplane
[2] ChatSplat: 3D Conversational Gaussian Splatting

**Other Strengths And Weaknesses:**

The paper is well-written and relative easy to follow.

NPISeg3D has a slightly larger parameter size than AGILE3D and Inter3D variants, which may restrict its application in scenarios with limited computing resources. While NPISeg3D provides uncertainty estimates, their reliability may need further verification and enhancement, especially with very few clicks.

**Questions For Authors:**

Reference to “Other Comments Or Suggestions”

**Relation To Broader Scientific Literature:**

The 3D interactive segmentation studied in this work can, to some extent, be used for constructing three-dimensional datasets in the medical field.

**Theoretical Claims:**

Most of them are reasonable, and I haven't checked all the formulas.

---

> ### Author Rebuttal · Authors · 2025-03-29
>
> ***Q1: Computational efficiency and Reliability of uncertainty estimation.***
>
> Thank you for your valuable comments. Below, we address each aspect in turn.
>
> ***Computational efficiency.*** Our  NPISeg3D introduces negligible extra parameters introduced by our neural process module which enhances generalization and enables reliable uncertainty modeling. As shown in Table 11 of the Appendix (pp. 14), NPISeg3D has 40.0MB of parameters, compared to 39.3MB in the previous SoTA AGILE3D. This amounts to only a 1.8% increase, which is minimal.
> **Despite this slight overhead, NPISeg3D delivers significantly improved segmentation performance**. For instance, It surpasses AGILE3D by 11.3% mIoU with 5 clicks under the single-object setting on KITTI-360, demonstrating that **the modest computational cost is well justified by the substantial performance gains**.
>
> ***Reliability of uncertainty estimation.*** To evaluate the reliability of our model’s uncertainty estimation, we provide extensive qualitative results in Figures 6 and 7 of the supplementary material. In particular, the first two rows of Figure 6 show that **after a single click, the uncertainty map effectively highlights erroneous regions and object boundaries—areas that are inherently ambiguous in interactive segmentation**. These results demonstrate that our method produces meaningful and reliable uncertainty estimates even with very few clicks. Moreover, as the number of clicks increases, we observe a clear reduction in uncertainty alongside improved segmentation quality, further validating the robustness and interpretability of the predicted uncertainty.
>
> ***Q2: Extension to other 3D representations, such as 3DGS. And potential integration with open-vocabulary 3DGS semantic segmentation methods, such as GOI and Chatsplat.***
>
> We appreciate the reviewer’s insightful comment. **Our current method is primary designed for point cloud inputs**, which provide a simple and efficient representation that generalizes well across diverse scenes without requiring scene-specific optimization.
>
> Meanwhile, we also agree that extending interaction to other 3D representations, such as 3D Gaussian Splatting (3DGS), is a highly promising direction. However, direct interactive segmentation in 3DGS is non-trivial: individual Gaussians lack explicit semantic meaning and are optimized per scene, making object-level interaction challenging. Recent works, such as Click-Gaussian [1], GaussianCut [2], and ISRF [3], have explored propagating 2D user interactions into 3D via dense multi-view supervision. These approaches typically rely on 2D-view-based interactions and project multi-view masks into 3D space.
>
> Inspired by these methods, **our method could be extended to such settings by first generating 2D masks from user clicks using our probabilistic framework, i.e., hierarchical neural process structure, then lifting these masks into 3DGS space using known camera poses or depth maps**. This could enable selecting or reweighting Gaussians based on segmented regions, facilitating interactive segmentation without requiring per-Gaussian semantic labels or retraining.
>
> Regarding integration with open-vocabulary 3DGS segmentation, methods such as GOI and ChatSplat incorporate language-guided semantics into 3DGS. We believe combining interactive inputs (e.g., clicks or referring expressions) with open-vocabulary reasoning is a compelling future direction. For instance, **user interactions could guide the adaptation of semantic hyperplanes or modulate Gaussian importance weights during inference**.
>
> **Although 3DGS and other 3D representations are not the main focus of our current work, we will include the above discussions in the final version to provide a broader perspective.** We also find it an interesting direction to explore how our probabilistic interactive framework could be adapted to 3DGS pipelines to support language-driven 3D interaction in future work.
>
> [1] Choi, Seokhun, et al. "Click-gaussian: Interactive segmentation to any 3d gaussians." European Conference on Computer Vision. Cham: Springer Nature Switzerland, 2024.
>
> [2] Jain, Umangi, Ashkan Mirzaei, and Igor Gilitschenski. "GaussianCut: Interactive segmentation via graph cut for 3D Gaussian Splatting." The Thirty-eighth Annual Conference on Neural Information Processing Systems. 2024.
>
> [3] Goel, Rahul, et al. "Interactive segmentation of radiance fields." Proceedings of the IEEE/CVF Conference on Computer Vision and Pattern Recognition. 2023.

---

### Official Review · Reviewer_F92a · 2025-03-14

**Overall Recommendation:** 3

**Summary:**

This paper proposes a method using neural processes for 3D interactive segmentation which in addition to segmentations, also enables uncertainty estimations. The proposed method uses a hierarchical latent structure to capture both local and global concepts and a probabilistic prototype modulator which allows for the model to have better object-aware context for its predictions. The paper validates its claims by comparing its method to other interactive 3D segmentation approaches showing improved performance for both single and multi-object segmentation and demonstrating improved generalization abilities over existing methods. Additionally, thorough ablations validate the importance of key method components (such as the hierarchical latent variables and the uncertainty estimation).

## Update after rebuttal
After reading the rebuttal, my concerns regarding further qualitative comparisons were addressed as more examples are shown in the supplemental material. However, I still think the paper would benefit form showing results on more scenes and more diverse scenes which could perhaps be added to the supplemental material. My other concern regarding generalization was addressed by the fine-tuning metrics presented in the rebuttal. While these results are convincing, it would help to include more detail on how much compute this fine-tuning requires. Additionally, it is not clear how feasible fine-tuning will be in real would applications. Having read the rebuttal, I am maintaining my score of weak accept.

**Claims And Evidence:**

The claims made are supported by sufficient evidence.

**Essential References Not Discussed:**

While not strictly necessary the paper might benefit from discussion of some interactive 3D segmentation methods that translate SAM features into 3D such as SAM3D[2] and SA3D[3].

References:
[1] Kirillov, Alexander, et al. "Segment anything." Proceedings of the IEEE/CVF international conference on computer vision. 2023.
[2] Yang, Yunhan, et al. "Sam3d: Segment anything in 3d scenes." arXiv preprint arXiv:2306.03908 (2023).
[3] Cen, Jiazhong, et al. "Segment anything in 3d with nerfs." Advances in Neural Information Processing Systems 36 (2023): 25971-25990.

**Experimental Designs Or Analyses:**

The experimental design appears sound.

**Methods And Evaluation Criteria:**

The methods and evaluation criteria proposed make sense for the given task. The authors evaluate their method on multiple segmentation datasets for both single and multiple objects as well as provide results as compared to user generated segmentations.

**Other Comments Or Suggestions:**

N/A

**Other Strengths And Weaknesses:**

Strengths:
- Improved performance over existing methods for interactive 3D segmentation
- Enables uncertainty predictions by using NP framework, a capability that is not supported by existing methods for this task

Weaknesses:
- Not enough qualitative comparisons, the paper would benefit from showing more qualitative results on diverse inputs. Perhaps this could be included in supplementary material if there is not enough space in the main paper.
- As discussed in the paper, while the generalization shown is an improvement over existing methods, it is not necessarily successful enough on challenging datasets such as KITTI-360 to be useful for downstream tasks.

**Questions For Authors:**

N/A

**Relation To Broader Scientific Literature:**

This paper approaches the task of interactive 3D segmentation using NPs and predicts the segmentations in a probabilistic manner enabling uncertainty predictions which are not supported by existing interactive 3D segmentation methods such as InterObject3D.

**Theoretical Claims:**

N/A

---

> ### Author Rebuttal · Authors · 2025-03-29
>
> ***Q1: Include discussion of some interactive 3D segmentation methods that translate SAM features into 3D such as SAM3D and SA3D.***
>
> Thank you for suggesting these relevant interactive 3D segmentation works, SAM3D and SA3D, both of which explore translating 2D SAM features into 3D. Specifically, SAM3D leverages the pretrained SAM to automatically generate 2D masks in RGB images, and then maps these masks into 3D point cloud using pixel-wise depth from RGB-D images—without requiring additional training or fine-tuning. Similarly, SA3D utilizes radiance fields as an off-the-shelf prior to bridge multi-view 2D images and 3D space. It first generates a 2D mask using SAM on a single view, and then performs mask inverse rendering and cross-view self-prompting to iteratively refine the 3D mask of the target object across multiple views.
>
> In contrast, our approach directly tackles native 3D interactive segmentation by taking 3D interactive clicks and point cloud inputs without relying on 2D projections. We view SAM3D, SA3D, and our method as complementary contributions that collectively advance 3D interactive segmentation.
>
> ***We will incorporate a discussion of these works in the final version*** to provide a more comprehensive perspective on related approaches.
>
> ***Q2: Supplementary materials issues.***
>
> We kindly clarify that the ***Appendix (supplementary materials) was included in the submission (pp. 12–20)***. In Appendix, we provided the ELBO derivation (Sec. A), additional quantitative and qualitative results (Sec. B) to show the effectiveness of our method across different settings, user study details (Sec. C) to demonstrate the usability in practical scenarios, and implementation details (Sec. D) such as model structure and click simulation strategy.
>
> ***Q3: More qualitative results on diverse inputs.***
>
> Thanks for this valuable comment. ***In Appendix (pp. 12–20), we provided extensive qualitative comparisons in Figures 4–7, showcasing diverse input scans and click prompts.*** Specifically, Figure 4 illustrates segmentation masks generated with varying numbers of clicks in the single-object setting, while Figure 5 presents results for multi-object segmentation. Figures 6 and 7 further display segmentation masks and uncertainty maps across different click counts for single- and multi-object cases. These examples cover challenging scenarios involving varying object complexities and sparse user inputs, highlighting the robustness and generalization capability of our method.
>
> ***Q4: Performance falls short on challenging out-of-domain datasets like KITTI-360.***
>
> Thank you for your feedback. Similar to prior methods such as AGILE3D [1] and Inter3D [2], our model is trained exclusively on ScanNet and evaluated on out-of-domain datasets like KITTI-360 to evaluate generalization. The substantial domain gap between ScanNet (indoor RGB-D scenes) and KITTI-360 (outdoor LiDAR scenes) inevitably leads to suboptimal performance on KITTI-360. As a result, even SOTA method like AGILE3D achieves only 44.4% mIoU after 5 clicks under the single-object setting.
>
> Nevertheless, our method demonstrates significant improvements over previous SOTA approaches. Specifically, our NPISeg3D improves mIoU by 10.9% and 9.8% over the previous SoTA AGILE3D on KITTI-360 under the single-object and multi-object settings, respectively, highlighting its effectiveness even under large distribution shifts.
>
> |               | **mIoU@5** | **mIoU@10** | **mIoU@15** | **NoC@80** | **NoC@85** | **NoC@90** |
> |---------------|------------|-------------|-------------|------------|------------|------------|
> | **w/o fine-tuning** | 44.0       | 48.5        | 52.9        | 16.4       | 17.0       | 17.6       |
> | **w/ fine-tuning**  | 79.2       | 82.8        | 85.4        | 8.5        | 10.8       | 12.3       |
>
> To enable effective deployment in downstream tasks, one feasible solution is to consider domain adaption techniques, e.g, domain-specific fine-tuning. As shown in the table above, our model—when fine-tuned on KITTI-360—achieves significantly better performance across all metrics, approaching levels that are practical for downstream tasks. For example, mIoU@5 improves from 44.0% to 79.2%, and the number of clicks (NoC@90) required to reach 90% IoU decreases from 17.6 to 12.3. These results demonstrate the strong adaptability of our method to specific domains when fine-tuning data is available.

---

### Decision · Program_Chairs · 2025-05-01

**Decision:**

Accept (poster)

**Comment:**

All reviewers agree to accept the papers. The author's rebuttal addressed reviewers' concerns, such as applications in other 3D representations like 3DGS. Please include all necessary changes in the final version.